



# Stable accumulation patterns around Dome C, East Antarctica, over the last glacial cycle

Marie G.P. Cavitte[1,2], Frédéric Parrenin[3], Catherine Ritz[3], Duncan A. Young[1], Donald D. Blankenship[1,2], Massimo Frezzotti[4], and Jason. L. Roberts[5,6]

[1]Institute for Geophysics, Jackson School of Geosciences, University of Texas at Austin, Austin, Texas, USA
[2]Department of Geological Sciences, Jackson School of Geosciences, University of Texas at Austin, Austin, Texas, USA
[3]Univ. Grenoble Alpes, CNRS, IRD, IGE, F-38000 Grenoble, France
[4]ENEA, Agenzia Nazionale per le nuove tecnologie, l'energia e lo sviluppo sostenibile, Rome, Italy
[5]Australian Antarctic Division, Kingston, Tasmania 7050, Australia
[6]Antarctic Climate & Ecosystems Cooperative Research Centre, University of Tasmania, Hobart, Tasmania 7001, Australia

*Correspondence to:* Marie G.P. Cavitte (mariecavitte@gmail.com)

**Abstract.** We reconstruct the pattern of surface accumulation in the region around Dome C, East Antarctica, through the last glacial cycle. We use a set of internal isochrones interpreted from various ice-penetrating radar surveys and a 1D pseudo-steady ice flow model to invert for both time-averaged accumulation rates and paleoaccumulation rates between isochrone pairs. We observe that

the surface accumulation pattern is stable through the last 128 kyrs, both the large-scale (100s km) gradients which reflect current modeled and observed precipitation gradients in the region, as well as the small-scale (10s km) accumulation variations linked to snow redistribution at the surface due to changes in its slope and curvature in the prevailing wind direction. This suggests a stable position of the dome throughout the last glacial cycle.

**1 Introduction**

The Dome C region, located on the East Antarctic interior plateau, has long been the focus of extensive research: it is the site of the oldest as-yet-retrieved ice core, the EPICA Dome C ice core, going back ~800 ka (Parrenin et al., 2007). It is also an area where surface precipitation is extremely low (Stenni et al., 2016). At other inland plateau sites (e.g. in Drönning Maud Land, Fujita et al., 2011),

occasional large precipitation events represent a large part (more than 50%) of the total annual precipitation, and this is the case at Dome C too (Frezzotti et al., 2005). Precipitation on the Dome C plateau is mostly dominated by coastal air masses which advect moisture inland. The presence of the dome creates an upslope and a downslope component of air flow: moisture is released preferentially



on the upslope/windward side of the dome, and the leeward/downslope side is therefore exposed to
drier air (Genthon et al., 2016). This is reflected in the large scale gradient of modern precipitation
measured (Arthern et al., 2006; Genthon et al., 2016; Kållberg et al., 2004) and modeled (Gallée
et al., 2013; Palerme et al., 2014; Van Wessem et al., 2014).

Although observations and model results suggest a spatially variable shift in dust particle sizes,
they indicate a uniform geographic provenance for mineral dust measured at EPICA Dome C. An
efficient and persistent westerly circulation pattern would have transferred dust from South America
and Australia to the East Antarctic plateau during glacial-interglacial cycles (Delmonte et al., 2010;
Albani et al., 2012). Present-day moisture-bearing air mass trajectories (Scarchilli et al., 2011; Gen-
thon et al., 2016) point to a western Indian Ocean provenance for the snow precipitation at Dome C
(85% of the precipitation), and suggest this could have persisted through glacial-interglacial cycles.
Snow precipitation is homogeneous at a large-scale, whereas local variations in snow accumulation
are controlled by local surface topography as a function of wind direction. Black and Budd (1964)
and Budd (1971) first observed the close relationship between bedrock relief, surface slope and accu-
mulation rates in Wilkes Land. Frezzotti et al. (2007) show that surface slope in the prevailing wind
direction (SPWD) is a key constraint in determining spatial and temporal variability of precipitation;
a higher SPWD can lead to significant ablation and redeposition of snow (Frezzotti et al., 2002b, a,
2005, 2007). Das et al. (2013) show that SPWD is a strong threshold for the formation of wind scour
or megadune fields.

Airborne and ground-based ice-penetrating radar data have long been used to constrain the surface
and bedrock topography over large parts of the Antarctic Ice Sheet (Gudmandsen, 1971; Drewry
et al., 1980; Millar, 1981; Siegert, 2003; Bingham and Siegert, 2009, and many others), as well
its internal stratigraphy (Siegert, 1999; MacGregor et al., 2012; Cavitte et al., 2016). Because the
internal stratigraphy represents isochronal surfaces throughout much of the ice sheet, dated internal
radar reflectors can be used to constrain the surface mass balance of the ice sheet (Medley et al.,
2013). Several radar isochrone studies have also shown the existence of a coast-to-dome precipitation
gradient: Verfaillie et al. (2012) show a continuous existence through historical timescales, while
Siegert (2003) shows the persistence of a strong accumulation gradient between Dome C and Ridge
B (a topographic high upstream of Lake Vostok) over glacial-interglacial timescales.

The position of Dome C and the adjacent ice divide have been presumed to be stable through time.
However, there is no reason that this should be the case: bounded to the east by the Byrd Glacier
catchment and to the west by the Totten Glacier catchment, these two glaciers have very different
flow behaviors due to different outflow and grounding line boundary conditions and therefore might
have influenced Dome C in an asymmetrical way. Shrinking of the stabilizing Ross Ice shelf on the
Byrd Glacier catchment side has been observed and modeled (Scherer et al., 1998; Conway et al.,
1999; Pollard et al., 2015), and Young et al. (2011) have shown evidence for significantly different
configurations of the Totten Glacier catchment over long time-scales. There is potential for the dome





to have migrated or disappeared over time to simply become part of the ice ridge if the glaciers destabilized at different times. Knowing the position of the dome is crucial for three reasons:

1. The position of topographic domes characterizes the spatial distribution of snow accumulation and the ice flow of outlet glaciers of the East Antarctic Ice Sheet (EAIS). Mass balance strongly affects sea level variations (past and future) as well as the geometry of the ice sheet through time. Several recent studies have shown the influence of increasing precipitation trends over Antarctica in a warming climate (Davis et al., 2005; Van Wessem et al., 2014; Frieler et al., 2015).

2. The position of the dome through time is required for accurate dating and interpretation of ice cores. Knowing the flowlines of ice particles through time is necessary to reconstruct ice core chronologies and correct for the effects associated with deposition at a different location and elevation than the ice coring site. Especially in the context of the search for 1.5 million-year-old ice, knowing the position of the dome through time will have a significant influence on the choice of an ice core site. Several candidate sites for such old ice have been identified in the region (Van Liefferinge and Pattyn, 2013).

3. The location of the dome is required to model isochrones interpreted from radar surveys. When modeling isochrones, the assumption that horizontal advection is negligible is only valid in close proximity to a dome or ice divide. If this is not the case, full 3D-type modeling with known dome and divide positions is necessary to reproduce isochrone geometries accurately (Parrenin et al., 2006; Leysinger Vieli et al., 2011).

Here, we reconstruct paleoaccumulation rates for the Dome C region using a 1D pseudo-steady ice flow model (described in the companion paper) for the last 128 kyrs using the isochronal constraints obtained from radar surveys. We discuss the large-scale accumulation and small-scale variations in accumulation which suggest a stable position of the dome for the last glacial cycle. We do not attempt to reconstruct older paleoaccumulations due to the 1D assumptions.

## 2 Methods

### 2.1 Dome C region

The Dome C region represents a topographic high in the middle of the EAIS and is at the confluence of several ice divides, the largest of which separates the Byrd Glacier catchment from the Totten Glacier catchment. The topography is gentle, reaching a maximum elevation at Dome C of ~3266 m above sea level (geoid height) where the change in elevation is ~10 m across 50 km (Genthon et al., 2016). A gentle saddle connects Dome C to Lake Vostok along the ice divide, with a secondary dome referred to as "Little Dome C" (LDC) just south of the Dome C ice core site. The bedrock is



characterized by a large subglacial massif ~40 km to the south of the Dome C ice core site and ~10
90  km south of the LDC, easily identifiable on Fig.1, where the radar survey grid is tightest. For ease of
description, we refer to it as the "Little Dome C massif" (LDCm) to differentiate from the surface
topographic high. The deep Concordia Subglacial Trench (CST) runs along its eastern edge and is
followed by a steep ridge, ~2000 meters high (Young et al., in review), which we will refer to as the
Concordia Ridge (CR). Both the LDCm and the CR (see Fig.1) have been identified as promising
95  targets for retrieving 1.5 million-year old ice (Van Liefferinge and Pattyn, 2013).

## 2.2 Radar data

We use several airborne ice-penetrating radar surveys collected in the Dome C region by the University of Texas at Austin Institute for Geophysics (UTIG) and the Australian Antarctic Division (AAD)
as part of the ICECAP project (International Collaborative Exploration of the Cryosphere through
100  Airborne Profiling, Cavitte et al., 2016) and the Oldest Ice candidate A (OIA) survey flown by ICE-
CAp in January 2016 (Young et al., in review) (Fig.1). All surveys use the same center frequency of
60 MHz; internal isochrones are therefore coherent from one season to the next. A set of 18 internal
isochrones are traced throughout the region, using the multiple crossovers, thus ensuring the reliability of the tracing as outlined in Cavitte et al. (2016). The co-location of the EPICA Dome C ice core
in the survey region enables the dating of the isochrones using the AICC2012 chronology (Bazin
et al., 2013; Veres et al., 2013). Obtaining ages and associated uncertainties for each isochrone is
described in Cavitte et al. (2016). We extend the same isochrones to the newly acquired OIA survey
and add a number of shallower and deeper isochrones in the OIA region (Cavitte et al., in prep.). We
use all 18 isochrones for the 1D model inversion but only use the youngest nine isochrones going
back to the penultimate interglacial (i.e. 128 ka) for paleoaccumulation reconstructions, explained
below. All nine isochrone depths, ages and uncertainties at the Dome C ice core site are given in
Table 1.

## 2.3 Modeling

We use 18 radar isochrones, dated from 10 ka (before 1950) to 366 ka, and the 1D pseudo-steady
ice flow model described in the companion paper (Parrenin et al., submitted). The model inverts for
time-averaged geothermal heat flux ($G_0$), time-averaged accumulation rate ($\bar{a}$), and time-averaged
vertical strain rate profile parameter ($p'$) every kilometer along a radar line. Pseudo-steady-state
means that all parameters in the model are considered steady except for $R(t)$, a temporal factor
applied to both basal melting and accumulation (see companion paper). In other words, we can split
the accumulation rate into a time-averaged component $\bar{a}(x)$ that varies spatially, and a temporally
varying component, $R(t)$:

$$a(x,t) = \bar{a}(x)R(t) \qquad (1)$$



$\bar{a}(x)$ therefore is the time-averaged accumulation rate at a certain point $x$, while $R(t)$ represents the variations in accumulation rate over glacial-interglacial cycles over time. The model assumes that $R(t)$ is spatially invariant over the entire study region. $R(t)$ is obtained from AICC2012 inferred accumulation variations (Veres et al., 2013; Bazin et al., 2013), and represents the ratio of the accumulation at time $t$ to the average accumulation over the last 800 kyrs.

When inverting the radar isochrones using the pseudo-steady ice flow model, ages and accumulations are all used in steady-state form, with glacial-interglacial accumulation variations normalized. The calculated time-averaged accumulation rate $\bar{a}$ (Fig.3), $p'$, and $G_0$ result from the best fit of all the radar isochrone depths (dropping $x$ for simpler notation). However, some differences between modeled and observed isochrones remain as all isochrones have to be simultaneously fitted for each point x. The 18 isochrones have to be used in the inversion as the deepest isochrones provide the strongest constraints on $p'$ and $G_0$.

To reconstruct paleoaccumulation rates through time $\bar{a}_{\Delta\chi}$, where $\Delta\chi$ represents a discrete age interval, we use the $G_0$ and $p'$ values calculated and assume they remain unchanged over each time so that the remaining misfit between modeled and observed isochrones is entirely a result of the uncertainty in $\bar{a}$. $\bar{a}_{\Delta\chi}$ represents the time-averaged paleoaccumulation rate for a layer with an age interval $\Delta\chi$, bounded above and below by a radar isochrone of AICC2012 age. We refer to these as isochrone-bounded layers. To calculate $\bar{a}_{\Delta\chi}$ values for each layer, we adjust the value of $\bar{a}$ such that modeled and observed isochrone-bounded layer age intervals $\Delta\chi$ are fitted perfectly for each layer.

In mathematical form, if $z$ is the depth of the isochrone and $\chi$ the age of the isochrone, we can write the isochrone-bounded layer's age interval as:

$$\Delta\chi_m = \frac{\Delta z}{\tau \bar{a}_{m,\Delta\chi}} \text{, for the model} \tag{2}$$

and

$$\Delta\chi_o = \frac{\Delta z}{\tau \bar{a}_{o,\Delta\chi}} \text{, for observations,} \tag{3}$$

where $\tau$ represents the vertical thinning rate. We therefore want to obtain $\bar{a}_{o,\Delta\chi}$, the "observed" paleoaccumulation rate for a certain age interval $\Delta\chi$.

Assuming all errors arise from accumulation rate uncertainty is equivalent to assuming $\tau$ is modeled perfectly. Therefore we can equate Eq.2 with Eq.3 and obtain $\bar{a}_{o,\Delta\chi}$:

$$\bar{a}_{o,\Delta\chi} = \frac{\Delta\chi_m}{\Delta\chi_o} \bar{a}_{m,\Delta\chi} \tag{4}$$

Using Eq.4, we calculate the best fit time-averaged paleoaccumulation rates through time in one iteration after the model inversion. The values of $\bar{a}_{o,\Delta\chi}$ obtained are the time-averaged paleoaccumulation for each isochrone-bounded layer of age interval $\Delta\chi$. This gives the spatial variations of the paleoaccumulation rates through time. Temporal variations $R(t)$ of the accumulation rates have


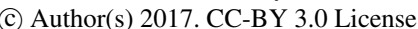


been ignored until this point. We use Eq.1 and calculated $R(t)$ values from the AICC2012 chronology accumulation variations to obtain the corresponding paleoaccumulation rates, $a_{o,\Delta\chi}$. These are shown in Fig.2 and 4.

In addition, the Metropolis-Hastings (MH) algorithm (described in the companion paper, Parrenin
et al., submitted) enables the calculation of an accumulation rate uncertainty which takes into account the age uncertainty of the radar isochrones. The age uncertainty of the radar isochrones is a combination of the radar depth uncertainties translated to age uncertainties (Cavitte et al., 2016) and the AICC12 ice core chronology uncertainties (Veres et al., 2013; Bazin et al., 2013). Cavitte et al. (2016) describe the various sources of radar depth uncertainty and how they are calculated. The radar
isochrone depth and age uncertainties are given in Table 1. We plot the time-averaged accumulation rate and the paleoaccumulation rates for each isochrone-bounded layer over the survey region (see Fig.3, 4). The accumulation uncertainties are given in Fig. S2.

Care must be taken in not over-interpreting the paleoaccumulation maps obtained. We do not argue that we have reconstructed absolute paleoaccumulations for the past 128 kyrs. The 1D pseudo-steady
ice flow model used here (see Parrenin et al., submitted) does not take horizontal advection into account. Instead, our paleoaccumulations are valid at the ice divide and the dome where horizontal ice flow speeds are negligible. Farther away, horizontal advection has a larger influence. We therefore focus on the OIA survey, which is closest to the dome, augmented only by radar lines from previous seasons nearest to the ice divide (Cavitte et al., 2016; Young et al., in review). Our paleoaccumulation
calculations do not apply deeper in the ice column where the assumption that $\tau$ (Eq.4) is fitted perfectly breaks down. We therefore reconstruct paleoaccumulation rates only over the last glacial cycle, and use the topmost nine isochrones which cover the period 10 - 128 ka (top half of the ice column). Furthermore, the model assumes a constant ice thickness through time. Even though small variations in the ice thickness through time will affect the absolute value of the reconstructed
accumulation rates, the assumption of constant ice thickness is fair for the center of the EAIS where modeled ice thickness variations have been reported up to 200 m (Bentley, 1999; Ritz et al., 2001) and is commonly assumed in ice core chronology reconstructions.

### 2.4 ECMWF ERA40 snow precipitation rate

The snow accumulation rates in the Dome C region result from precipitation in the form of snow
(snowfall and diamond dust), then modified by wind-driven processes. The wind erosion, wind redistribution and sublimation, as well as other processes during or after a precipitation event, leads to a spatial deposition at the surface that is much less homogeneous than the original precipitation (e.g., Frezzotti et al., 2004). To compare large-scale patterns of precipitation to independent measurements, ECMWF (European Center for Medium-Range Weather Forecasts) ERA40 re-analysis
data (Simmons et al., 2007) is used to obtain a map of present-day estimated precipitation rates over the survey region. The ECMWF ERA40 model seems to correctly reproduce the observed precipita-





tion's spatial and temporal variability at Dome C, but systematically underestimates the precipitation magnitudes (Genthon et al., 2016; Stenni et al., 2016), probably because clear-sky precipitation is not adequately parameterized (Bromwich et al., 2004; Van de Berg et al., 2006). The ECMWF ERA40 model does not reproduce snow accumulation because it does not consider the blowing snow transport/sublimation process. However, since the Dome C site is not influenced by strong winds, this is expected to have a minor effect within the summit area, but cannot be completely neglected farther than 25 km from the dome/ice divide. ECMWF ERA40 data have been normalized using the surface accumulation average of the last centuries from existing ground-penetrating radar (GPR) within 25 km from Dome C summit (Urbini et al., 2008).

A number of steps went into creating this data set, shown in Fig.5:

1. ECMWF ERA40 monthly average precipitation rates were used to calculate a long term precipitation average over the 1989 - 2011 period

2. Precipitations were then interpolated over the region of interest as a 1 km grid

3. Precipitation values were increased by 12.9 mm $yr^{-1}$ to match GPR measurements in the area (Urbini et al., 2008) as ECMWF ERA40 precipitation values are systematically too low compared to ground-based measurements.

Independent traverse accumulation measurements confirm the calculated accumulations (Emmanuel Le Meur, pers comm.)

### 2.5 Detrending paleoaccumulation rates

To look at small-scale paleoaccumulation variations more closely, we remove large-scale precipitation gradients (see Sect.4). For this, we calculate a quadratic fit of the ECMWF ERA40 surface accumulation values (as described above) with each isochrone-bounded layer's paleoaccumulation, and subtract the calculated fit from the layer's paleoaccumulation values. The result is a map of detrended paleoaccumulations for each isochrone-bounded layer (Fig.6).

### 2.6 Slope and Curvature in the Prevailing Wind Direction (SPWD and CPWD)

In Sect.4, we discuss the importance of surface slope in the prevailing wind direction (SPWD) and curvature in the prevailing wind direction (CPWD). We use ECMWF 5-year average wind directions (Simmons et al., 2007) and Bamber et al. (2009) surface elevations to calculate SPWD and CPWD values over a 3 km radius in the survey region (Fig.6). A positive value of surface curvature indicates a surface trough, while a negative value of surface curvature indicates a surface bump.



## 3 Results

We use a standard MH algorithm to run the pseudo-steady ice flow model to invert for time-averaged $\bar{a}$, $p'$ and $G_0$. Values of the time-averaged $\bar{a}$, $p'$ and $G_0$ and their uncertainties are obtained after 1000
MH iterations, each taking 5 thermo-mechanical iterations (see companion paper, Parrenin et al., submitted). Parrenin et al. (submitted) describe the results obtained and the parameter priors used for the inversion. Here, we focus on the accumulation rate reconstructions $\bar{a}$ and $a_{o,\Delta\chi}$ for each isochrone-bounded layer, obtained using Eq.4.

The reconstructed paleoaccumulations $a_{o,\Delta\chi}$ are shown in the top panel of Fig.2 along the A-
A' radar transect (VCD/JKB2g/DVD01a) marked on Fig.1. Ages given are the mean of the age interval represented in each paleoaccumulation rate. The A-A' radar line runs along the ice divide, and a marked decreasing gradient can be seen going from the northeast side towards the southwest consistently over all age intervals. Bottom panel of Figure 2 displays $a_{o,\Delta\chi}$ along the B-B' radar transect (OIA/JKB2n/Y77a) marked on Fig.1. This transect runs across the divide and there is no
clearly visible accumulation gradient over time for most isochrone-bounded layers, except a weak one for the interglacial 10 ka and 128 ka isochrones.

We also show reconstructed accumulation rates in map view in Fig.3 and 4. Fig.3 displays the time-averaged accumulation rate $\bar{a}$ and Fig.4 displays the paleoaccumulation rate per isochrone-bounded layer $a_{o,\Delta\chi}$. We show six of the age intervals calculated. We observe that the time-averaged
accumulation (Fig.3) has a clear north to south gradient, decreasing from $> 21$ mm water equivalent per year (mm-we yr$^{-1}$) in the north to 15 mm-we yr$^{-1}$ in the south. Superimposed, we observe a number of regions ~20 km wide that show a ~25% accumulation increase over the LDCm, to ~75% increase over the CR. These are outlined by black lines on Fig.3. Around the CR, we also note that the extended area of high accumulation is preceded by an area of very low accumulation, parallel to
it and just east of the CST. This corresponds to an area of drastic surface slope and curvature change (see also Fig.6, Sect.4 and S3).

Paleoaccumulation rates per isochrone-bounded layer (Fig.4) show a similar pattern in the accumulations: a large-scale gradient N-S with superimposed areas of higher accumulation in the same locations as in the time-averaged reconstruction. We note a striking similarity between the time-
averaged accumulation rate (Fig.3) and the paleoaccumulation rates for the ages 0 ka - 38 ka (Fig.4). We also note that gradients are stronger for interglacial age intervals (0 ka - 10 ka and 121 ka - 128 ka in Fig.4). The small-scale accumulation patterns remain spatially stable through time: panels in Fig.4 ranging from 0 ka to 106 ka display these same three areas of high accumulation outlined in Fig.3. The areas are ~20 kilometers wide and ~50 km or more east of the CR. However, they are less
prominent on the 121 ka - 128 ka panel on Fig.4, except for east of the CR.

Also plotted on Fig.3 and 4 are bedrock elevations from Bedmap2 (Fretwell et al., 2013) augmented with new OIA survey data outlined with a dashed rectangle (Young et al., in review), as well as Bamber et al. (2009) surface elevation contours . The areas of higher accumulation are co-located

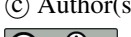



with areas of low surface slopes, visible from the surface contours. The accumulation variations we
observe are also co-located with significant bedrock relief changes, which reach e.g. ~2000 m for
the CR escarpment, and ~500 m for the south side of the LDCm (see Fig. S1).

To focus on the small-scale variations in paleoaccumulations, we use the time-averaged accumulation, $\bar{a}$, obtained from the model and Eq.1 to plot Holocene
average accumulation rates. For this, we take the ratio of the average accumulation rate of the last
100 years to that of the last 800 kyrs using the AICC2012 chronology, which has a value of 0.65.
The time-averaged $\bar{a}$ is scaled by this factor of 0.65 to obtain Holocene average accumulation rates,
which we call $a_{100yrs}$ here. We plot $a_{100yrs}$ together with ECMWF ERA40-based precipitation data
in Fig.5 (see Sect.2.4 for details). We observe that the large-scale N-S accumulation gradient in
$a_{100yrs}$ closely resembles that of the ECMWF ERA40-derived surface accumulation rate in Fig.5:
high accumulation in the north nearer the coast, and lower accumulation in the south as you move
towards the interior. The magnitude of the accumulation rates also match surprisingly well.

The calculated accumulation rate uncertainties from the model, with an average value of 0.16 mm-
we yr$^{-1}$ (see Fig. S2), are an order of magnitude (or more) smaller than the values of reconstructed
time-averaged accumulation rate, providing confidence in the time-averaged accumulation rates cal-
culated. However errors have been treated as uncorrelated so we cannot apply these uncertainties to
the paleoaccumulations. We hope to improve this in the future.

To focus on the small-scale variations in paleoaccumulations, we plot detrended paleoaccumula-
tions (Sect.2.5) for the region on top of SPWD and CPWD values (Sect.2.6), as shown on Fig.6. As
a reminder, these accumulation maps therefore display values of detrended paleoaccumulation once
the large-scale precipitation gradient obtained from ECMWF ERA40 has been removed. Looking
at the spatial distribution of these detrended paleoaccumulations in relation to SPWD, we observe
that areas with high accumulation are co-located with areas of markedly reduced SPWD values with
respect to the surrounding values (~0.5-1.2 x 10$^{-3}$ of absolute SPWD decrease). This is displayed
in Fig. S3. But more striking is the clear relationship between the magnitude of the curvature (and
polarity) and the magnitude of the residual paleoaccumulation (Fig.6). Areas of high positive de-
trended paleoaccumulation, $> 1.2$ mm-we yr$^{-1}$, are well correlated with areas of strongly positive
curvature values ($> 2$ x 10$^{-7}$ m$^{-1}$). This is evident in the LDCm area for the high accumulation
areas highlighted on Fig.3. Areas of high negative detrended accumulation, $< -1.6$ mm-we yr$^{-1}$, are
also well correlated with areas of strongly negative curvature. This is best seen east of the CR. Note
that the correlation holds particularly well for the youngest layer (0 - 10 ka); the residual paleoac-
cumulation values east of the CR are high directly where the surface curvature is strongly positive,
low where the surface curvature is strongly negative (blue on blue, red on red on Fig.6). However,
for layer 10 - 38 ka and older, this relationship is slightly offset, best visible east of the CR area.



## 4 Discussion

The observed patterns of paleoaccumulation agree well with previous studies of surface snow ac-
cumulation variability in the Dome C region. Considering first the large-scale patterns in the accu-
mulation reconstructions (Fig.3 and 4), we observe a consistent large-scale gradient for each age
interval. Large-scale here refers to 100s of kilometers. Accumulation decreases from the north side
of Dome C to the south side. This is clearly seen in ECMWF ERA40 data for the region (see Fig.5).
Other large scale accumulation models of the region (e.g. Genthon et al., 2016) or Regional Cli-
mate Model (MAR) (Gallée et al., 2013, 2015) also display such accumulation patterns. GPR data
collected during traverses across Dome C and along the divide also show a clear N-S gradient in
accumulation (Urbini et al., 2008; Verfaillie et al., 2012, Emmanuel Le Meur, pers. comm). A SPRI
airborne transect collected over Dome C also shows a strong accumulation gradient of 10s of mm
yr$^{-1}$ over a spatial scale of 100s of km (Siegert, 2003).

The fact that our paleoaccumulation reconstructions reproduce present-day surface accumulation
gradients and that this remains true back to 128 ka suggests both a stable meteorologic system and
location of Dome C. To preserve the same N-S gradient, the moisture-bearing air masses coming
from the coast must have interacted with the same surface topography during those 128 kyrs. Mea-
surements made in other areas of the ice sheet, e.g. across Talos Dome (Frezzotti et al., 2007), point
to similar patterns: accumulation is highest near the moisture source and decreases with distance
from the coast. Fujita et al. (2011) point to the same patterns of reduced accumulation inland across
Dronning Maud Land. Such a consistent large-scale depositional pattern indicates a stable regional
topography of the Dome C region. Urbini et al. (2008) show a small component of counter-clockwise
rotation of the accumulation pattern in historical times centered on Dome C, but the general N-S gra-
dient difference in accumulation across the dome remains. We note a good agreement between our
accumulation values and trends along A-A' going from Dome C along the ice divide towards Vostok
(top panel of Fig.2) and the GPR transect measured by Verfaillie et al. (2012) on the other side of
the Dome C divide.

Considering the small-scale (10s of kilometers or a few ice thicknesses) patterns of accumulation
shown earlier, we described several regions of locally higher accumulation. The co-location of the
areas of higher accumulation with areas where surface slope is reduced, as seen from the surface
contours or the markedly reduced SPWD values with respect to the surroundings (Fig. S3) fits well
with the model put forward by Frezzotti et al. (2007) over Talos Dome where accumulation increases
when SPWD decreases. They attribute the correlation between the absolute magnitude of SPWD and
accumulation rates to katabatic wind-driven ablation. Note that the prevailing wind direction over
the area is more or less along the long axis of Dome C flowing from higher up the ice divide towards
Dome C (Frezzotti et al., 2005; Urbini et al., 2008).

The spatial correlation we obtain between the detrended paleoaccumulations and the CPWD can
be explained by the same mechanisms as for SPWD, since SPWD and CPWD are directly related.



The proximity of the isochrone-bounded layer to the surface influences how well the correlation holds, particularly visible in the CR region. Layer 0 - 10 ka shows high detrended paleoaccumulation values where the surface curvature is strongly positive (i.e. surface trough), and low values where the surface curvature is strongly negative (i.e. surface bump). For any deeper layer, this relationship is slightly offset in space. This could be due to ice flow increasing with distance from the dome. Accumulation rates are calculated using radar isochrones at depths between ~9% and 53% of the ice thickness at Dome C. Therefore, we expect that a strongly positive CPWD region would affect the accumulation rate directly above this same region. These anomalies are preserved by progressive burial and as ice flow speed over the CR relief is higher than over the LDCm, radar isochrones observed at the CR will have carried this anomalous accumulation signature further down-flow than they would have at the LDCm.

Even though the absolute magnitudes of slope and curvature changes we observe are relatively small (on the order of $10^{-3}$ and $10^{-7}$ m$^{-1}$, respectively), other studies have shown that even very small slope changes can have a strong influence on wind-borne redistribution of snow (Grima et al., 2014; King et al., 2004). However a single mechanism has yet to be described that would explain the relationship between CPWD (and therefore SPWD) and small-scale accumulation variations. Grima et al. (2014) observe strong surface density variations linked to surface slope breaks, however some increases in accumulation occur over steeper surface slopes, which is surprising when steep slopes are usually associated with reduced accumulation (Hamilton, 2004; Frezzotti et al., 2004). King et al. (2004) show that local slope changes of 0.01 can create up to 30% variations in accumulation, and invoke a highly non-linear relationship between wind speed and snow transport to explain the type of accumulation variability they observe. We attempted a series of low order (linear and quadratic) fits between CPWD and our detrended paleoaccumulations but none explain all the variability. The data is suggestive of threshold behaviors between low and high CPWD magnitudes. ECMWF wind speed magnitudes over the LDCm and CR areas (Simmons et al., 2007) are below the 5 m s$^{-1}$ threshold for dune processes to be active in the region, and the radar data used does not show any buried dune structures. The accumulation patterns observed are more suggestive of the preferential infill of surface troughs by winds. These troughs might not fill-up easily because of the very low surface precipitation rates in the region (Genthon et al., 2016; Urbini et al., 2008) combined with the presence of areas of subglacial melting in the region (Young et al., in review), creating additional draw-down of the surface.

Although we cannot yet explain the mechanisms causing the small-scale paleoaccumulation variability we observe in the Dome C region, our results have important ramifications for constraining the region's stability through time. If we assume slope morphology in the prevailing wind direction is the dominating control on accumulation variability, the temporal persistence of the patterns of accumulation is only possible if the surface morphology, i.e the SPWD and CPWD, is also spatially unchanging, independent of whether the control comes from the bedrock topography. If the surface



slope and curvature do not change, we can suppose this implies the position of the divide and the dome must have remained relatively stable. This implies that the present-day configuration of the ice sheet in the region could have been the same through the last 128 kyrs. Geometric information on
Dome C and its surrounding region is critical and so far unconstrained in climate reconstructions of the EAIS. Dome C and the surrounding ice divides have long been modeled as stable spatially but this hypothesis has lacked evidence. The results shown here provide the first piece of such evidence.

Note the extreme pattern of high and low accumulation parallel to the CST and east of the CR seems to be the ideal example of how surface topography variations affect accumulation rates. The
ice flowing radially away from Dome C has to flow over CST and over the prominent bedrock CR. CPWD shows strongly negative values over the subglacial CR; it creates a surface which is concave down perpendicular to the wind direction. We can imagine a scenario in which snow is strongly plucked away on this steepest surface slope, but further down-wind, as slope reduces and reaches contrastingly strongly positive CPWD, the snow can then be redeposited directly down-wind as
suggested in Frezzotti et al. (2004).

We noted in the results that the small-scale accumulation variations were co-located with bedrock relief variations (see Fig. S1). Frezzotti et al. (2007) explain that bedrock topography can be the underlying influence on the variability of snow accumulation at scales of 1-20 km, corresponding to the lengthscales of the accumulation variations we calculate here. Bedrock topography will have a
stronger influence on the overlying ice in the presence of subglacial lubrication (Rémy et al., 2003). Rémy et al. (2003) show that for the Dome C region the most positive surface curvatures are directly linked to the largest ice thicknesses and the presence of subglacial lakes. It is interesting to note that areas of higher detrended paleoaccumulation correlated to high positive CPWD in this study are above deep bedrock valleys dotted with many observed subglacial lakes (Young et al., in review).
One interesting area is the band of high positive curvature highlighted on Fig.6 with a black arrow. Here we observe a band of high detrended paleoaccumulation between 0 - 73 ka intervals, but not in the 73 - 82 ka interval. It seems instead to get displaced further west. Ice here is resting in a shallower valley with limited subglacial lakes. The difference in behavior could be linked to transient freezing of the bedrock in regions of thinner ice which would lead to transient surface topography
Small-scale accumulation patterns appear to be different for the age interval 121 - 128 ka. This period represents the penultimate interglacial and it is possible surface topography changes occurred in the Dome C region. However, we do not attempt to further explain these changes because this layer is at a depth of ~50% of the ice column where the 1D inversion is less robust. This will be testable with a 3D model.
More accurate absolute accumulation rates could be obtained using a full 3D thermo-mechanical model. Further GPR data was recently collected over the LDCm, and strain nets and various other instruments were deployed. These new measurements will allow a test of the accumulation recon- struction of this study.



## 5 Conclusions

We reconstructed accumulation rates for the last 128 kyrs. Looking at both large- and small-scale accumulation gradients, we show that these have not changed significantly over the last glacial cycle. Large-scale accumulation gradients will remain constant if moisture-bearing air mass trajectories do not vary, which means that the topographic controls must remain unchanged. Small-scale accumulation variations are strongly controlled by SPWD and CPWD and therefore, if the pattern of high

and low accumulations remains fixed over a long period of time, this requires consistent interactions between local surface slopes and prevailing winds over the last 128 kyrs, independent of whether the control comes from the bedrock topography and/or potential basal melting. Both suggest that the current surface topography of the Dome C region has not changed significantly over the last glacial cycle. This points to a stable position of the Dome C region and adjacent ice divides, an impor-

tant constraint for modeling efforts in the area, both for dating existing ice cores as well as for the prospecting of a greater than 1 million-year-old ice core site.

## 6 Data accessibility

The radar isochrones used in this manuscript will be made publicly available in summer 2017 as a separate publication. Code for the model is available publicly under https://github.com/parrenin/IsoInv.

## 7 Author contributions

M.G.P. Cavitte interpreted the radar isochrones, F. Parrenin developed the model and ran experiments with M.G.P. Cavitte with C. Ritz input, D.A.Young, J.L Roberts and D.D. Blankenship were involved in survey design and data acquisition, M. Frezzotti provided data and discussion material. M.G.P. Cavitte prepared the manuscript with contributions from all co-authors. The authors declare that they

have no conflicts of interest.

*Acknowledgements.* This research was made possible by the joint French–Italian Concordia Program, which established and runs the permanent station Concordia at Dome C. This work was supported by NSF grants ANT-0733025 and ARC-0941678, NASA grants NNX08AN68G, NNX09AR52G, and NNX11AD33G (Operation Ice Bridge) to Texas, the Jackson School of Geosciences, the Gale White UTIG Fellowship, the G. Unger Vetle-

sen Foundation, NERC grant NE/D003733/1, the Global Innovation Initiative award from the British Council, the Australian Government's Cooperative Research Centre's Programme through the Antarctic Climate and Ecosystems Cooperative Research Centre (ACE CRC). Operational support was provided by the U. S. Antarctic Program and by the Institut Polaire Français Paul Emile Victor (IPEV) and the Italian Antarctic Program (PNRA and ENEA) and the Australian Antarctic Division provided funding and logistical support (AAS 3103,

4077, 4346). We acknowledge the support of Kenn Borek Airlines. Additional support was provided by the





French ANR Dome A project (ANR-07-BLAN-0125). Special thanks to Olivier Passalacqua for fruitful discussions. This is UTIG contribution 3116.



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





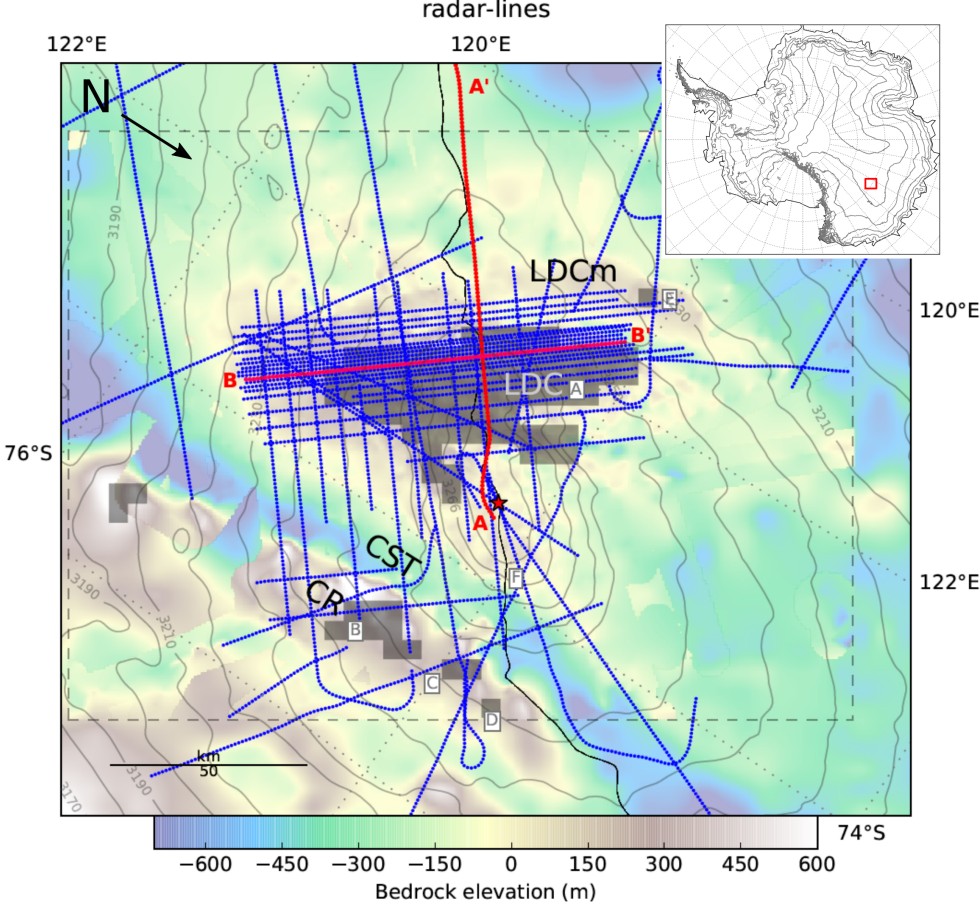

Figure 1: Map of Dome C and the surrounding region. A red square locates the study area on the inset. The radar lines used in the accumulation reconstructions are displayed as blue lines. Highlighted in red are the two radar lines shown in Fig.2. Dark gray blocks labeled A-E are the Van Liefferinge and Pattyn (2013) Candidate regions. F labels a 1.5 million-year-old ice new Candidate site (see companion paper). The background is bedrock elevation in meters above sea level and combines Bedmap2 bed elevations (Fretwell et al., 2013) as well as a recompilation based on the OIA radar bed elevations (Young et al., in review) delimited by a dashed rectangle. Gray lines are Bamber et al. (2009) surface elevations, a black line locates the ice divide. A red star locates the EPICA Dome C ice core. LDC locates the gentle secondary surface dome, LDCm locates the Little Dome C massif under the densest radar lines, CR locates the Concordia Ridge steep escarpment along the Concordia Subglacial Trench (CST).



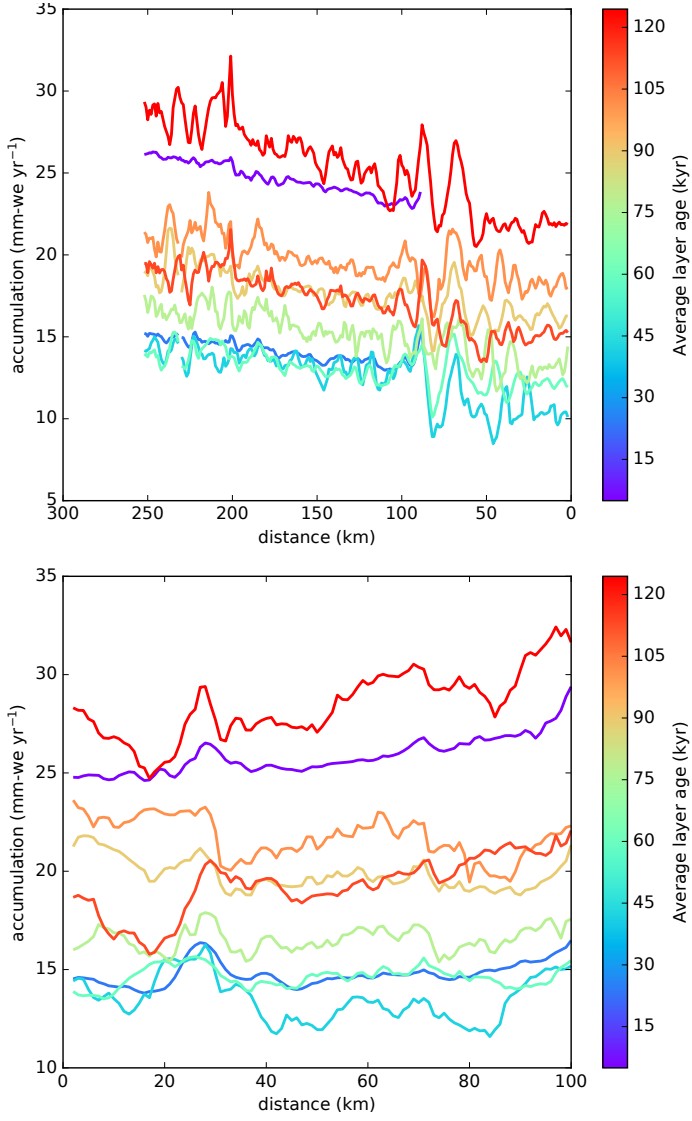

Figure 2: Paleoaccumulation rates along radar lines. Colors represent the mean of the age interval $\Delta\chi$ represented by each layer. Top panel shows the reconstructed paleoaccumulation rate $a_{o,\Delta\chi}$ along the A-A' radar line. Bottom panel is along the B-B' radar line. Both radar lines are highlighted on Fig.1. A-A', along the ice divide, displays a strong and consistent accumulation gradient. B-B', perpendicular to the ice divide, shows no similar gradient.

Figure 3: Time-averaged accumulation rates $\bar{a}$ along the radar lines over the Dome C region. Accumulation rates are given in mm of water equivalent per year. There is a clear large-scale N-S accumulation gradient, with accumulation decreasing with distance from the Indian Ocean coast, the main pathway of snow precipitation. Black lines outline areas of small-scale high accumulation: they correlate to areas where surface contours (in gray) become further apart, i.e. where surface slope is reduced. Background is the same as in Fig.1.










Figure 4: Paleoaccumulation reconstruction over the Dome C region. Panels show paleoaccumulation rates calculated for each isochrone-bounded layer, age intervals are given on each panel. The N-S accumulation gradient decreasing with distance from the Indian Ocean coast and the small-scale areas of high accumulation both remain stable through the last 128 kyrs. Small-scale accumulation variations are less visible for 121-128 ka. Background is the same as in Fig.1.



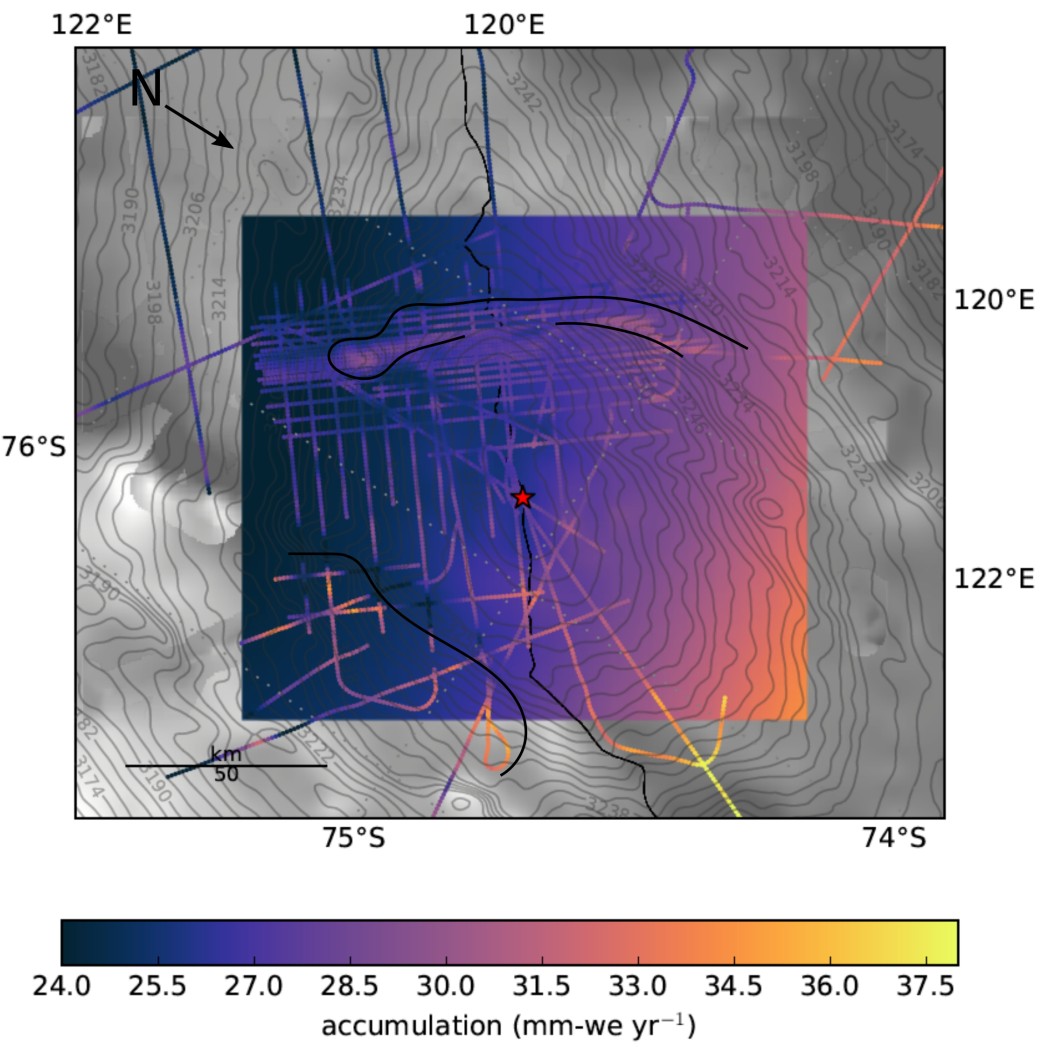

Figure 5: Holocene average accumulation rates $a_{100yrs}$ along the radar lines superimposed on ECMWF ERA40 estimated present-day surface accumulation rates (see Sect.2.4). There is a very good agreement in the magnitude of accumulation values between the two datasets and in their N-S accumulation gradient on large-scales (100s km), with accumulation decreasing with distance from the coast. White lines outline the same areas of small-scale high accumulation as in Fig.3. Background is the same as in Fig.1.

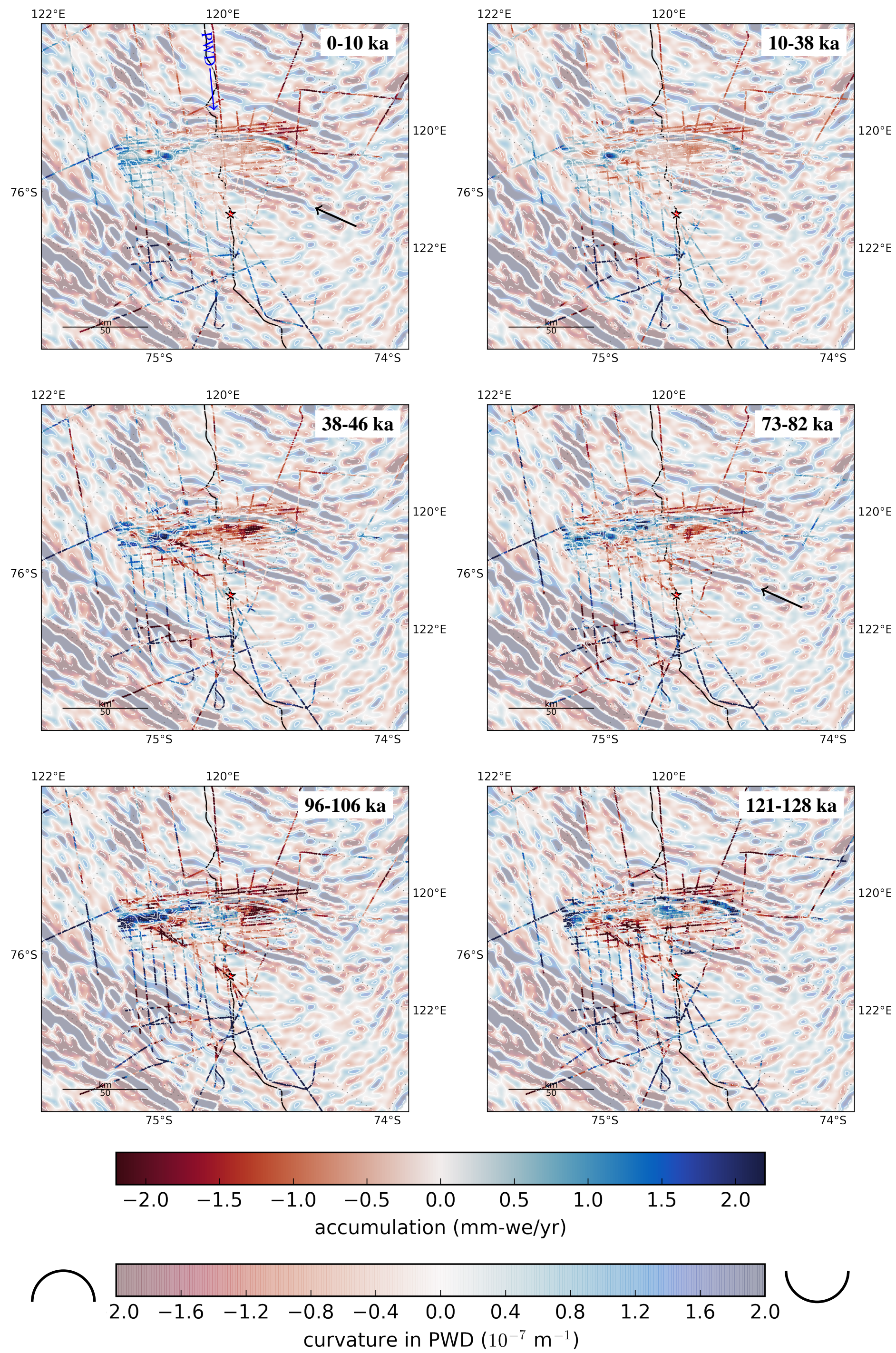
0-10 ka
10-38 ka
38-46 ka
73-82 ka
96-106 ka
121-128 ka
122°E
120°E
76°S
75°S
74°S
km
50
accumulation (mm-we/yr)
−2.0 −1.5 −1.0 −0.5 0.0 0.5 1.0 1.5 2.0
curvature in PWD ($10^{-7}$ m$^{-1}$)
2.0 −1.6 −1.2 −0.8 −0.4 0.0 0.4 0.8 1.2 1.6 2.0




Figure 6: Residual paleoaccumulations over the region, overlain on surface curvature in the prevailing wind direction (CPWD, strongly positive and negative values are sketched on either end of the colorbar). Panels show the same age intervals as Fig.4. The residual paleoaccumulation highs correlate well to areas of strongly positive CPWD. As layers get older, increasing offsets are visible east of the CR. A black arrow indicates the area of surface curvature discussed. A blue arrow indicates prevailing wind direction.

Table 1: Radar isochrones and their uncertainties at the Dome C ice core site.

| Isochrone | Depth (m) | Depth uncertainty (±m) | Age (ka) | Age uncertainty (±ka) |
|---|---|---|---|---|
| 1 | 307.61 | 1.82 | 9.97 | 0.26 |
| 2 | 699.60 | 2.29 | 38.11 | 0.61 |
| 3 | 798.60 | 2.31 | 46.41 | 0.80 |
| 4 | 1076.10 | 3.11 | 73.37 | 2.07 |
| 5 | 1171.90 | 3.18 | 82.01 | 1.55 |
| 6 | 1337.90 | 3.78 | 96.49 | 1.74 |
| 7 | 1446.80 | 3.97 | 106.25 | 1.83 |
| 8 | 1593.90 | 4.32 | 121.09 | 1.70 |
| 9 | 1682.10 | 4.51 | 127.78 | 1.78 |