# Peer review of "Accumulation patterns around Dome C, East Antarctica, in the last 73 kyrs"

_The Cryosphere, 2017_

## Referee Comment (RC1) · Anonymous Referee #1 · 30 Jun 2017

Working toward a better understanding of the ice-dynamic history around Dome C is an important pursuit, and the idea of using observed isochrones to better constrain divide dynamics is a good one. However, this manuscript is missing some critical references which highlight both the challenges of this type of work and the requirements for any model used to derive isochrone position near an ice divide. To make this manuscript a significant contribution to the literature, the authors need to better justify their model in the context of previous work, and define a more precise set of conclusions.

A full review is attached.

Please also note the supplement to this comment:
https://www.the-cryosphere-discuss.net/tc-2017-71/tc-2017-71-RC1-supplement.pdf

[Figure]

[Figure]

**Supplement:**

**Review Comments**

Title:Stable accumulation patterns around Dome C, East Antarctica, over the last glacial cycleJournal:The CryosphereDate:06/30/2017

**General Comments:**

In this paper, the authors use the spatial pattern of model derived paleo-accumulation rates [generated as part of a companion study (Parrenin et al. 2017)] to constrain the processes controlling surface mass balance around Dome C, East Antarctica, and infer the long-term stability of the ice divide presently located there. A 1D model inversion computes three free parameters – the average accumulation rate, the average geothermal flux, and the average velocity shape function – given temporal variability in accumulation set by the time-series derived from the EPICA Dome-C core. Using the geothermal flux and velocity shape function from that inversion, the authors then recompute the time-history for the accumulation rate to perfectly match the observed isochrons. From this accumulation rate product, the authors make the claim that the "surface topography of the Dome C regions has not changed significantly over the last glacial cycle [128 ka]."

There is a history of researchers using modeled isochron position to determine paleoaccumulation rates near several of the deep ice cores in Antarctica and Greenland, most notably at WAIS Divide (e.g., Koutnik et al. 2016; Neumann et al. 2008) and NGRIP (e.g., Fahnestock et al. 2001). Both this study and the companion paper are missing all reference to this body of work, which, if read, would have highlighted challenges associated with deriving a unique solution for accumulation rates that go undiscussed in the current manuscript. Most notably, I think the authors of the previous works would take issue with the following model choice, stated in the companion paper:

"We use a 1D pseudo-steady (Parrenin et al., 2006) ice flow model, which assumes that the geometry, the shape of the vertical velocity profile, the ratio  $\mu=m/a$  and the relative density profile are constant in time." (Line 104-105, (Parrenin et al. 2017))

As seen from the work at WAIS divide (Koutnik et al. 2016), the accumulation history trades-off nonuniquely with the local horizontal stretching rate and burial history (which is dependent on distance from divide), so by holding their "p" shape value constant in time and ignoring horizontal advection, the authors of this work *prescribe* a result with no changes in flow regime (ie, unchanging distance to the divide) at every position. This makes it impossible to constrain the actual stability with their results as presented, as their conclusion that the divide is stable is baked into their model assumptions.

In the absence of the ability to prove that the Dome C divide has been unchanging for 128 ka, I find it difficult to see the novel contribution of this paper.

**Specific Comments:**

The primary limitation of this paper is in the model formulation (which itself is under-review, making this manuscript harder to evaluate in isolation). Previous literature has documented a more thorough approach

for this research question. The analysis of isochrons connected to the WAIS divide ice core included an exhaustive exploration of model space, varying both divide position and flow dynamics. Even over a 9.2 ka reconstruction, non-uniqueness in the WAIS divide solution made it difficult to make claims about accumulation and divide position with high certainty. Constraining results over 128 ka will be proportionally more difficult. To reach the level of rigor demonstrated in previous studies, this work needs to do the following:

**1) Define "stable" -**

This manuscript describes the accumulation pattern and divide position as "relatively stable" several times, without any context for what that term means. Migrates or evolves at rates less than X? Has not moved beyond a range greater than X over time period Y? There is a general lack of precision in both the discussion and conclusions that must be refined.

**2) Clearly establish criteria for evaluating stability -**

In the current draft, the authors have not done enough to identify *how* they have inferred divide stability. My reading of the draft shows them making two claims: (1) spatial patterns in small-scale accumulation variability have been persistent through time, and (2) large scale accumulation gradients consistently show lower accumulation to the South. Without getting into the accuracy of the model produced accumulation rates, it is not clear that either of these conclusions justify an inference of divide stability. The authors ascribe small scale surface variability (which controls local mass balance) to processes at the bed, which means their temporal stability is not at all controlled by divide position. The large-scale accumulation pattern seen today depends mostly on relative distance to the coast. Without a clear marker for divide position in the current accumulation field, along with a demonstration that such a transition has not migrated with time, information in the large-scale gradient also seems insufficient to prove divide position has not changed.

**3) Evaluate divide stability in the context of other evidence –**

A perfectly stable divide over the last 128 ka would develop a prominent, measurable Raymond arch. Given no Raymond arch presented, my suspicion is that there is not one under the current divide, which should provide evidence for at least some instability in the system. Given this observation, the authors could provide a bound for *the most* stable the divide could be.

**4) Explore the possibility of a temporally variable divide position with your model –**

This represents one of the biggest hurdles for the authors going forward. As written, this inverse model formulation is not suitably justified. This is in part because the model development is established in the companion paper, but the objectives of that paper and this one are quite different. In (Parrenin et al. 2017), the goal is to constrain the age of the deep ice. For that work, trade-offs in the strain thinning and accumulation rates do not matter, as their *combined* effect dictates the age of the ice. For the inferences you try and make in this paper, you need a unique solution that disentangles ice-flow effects from accumulation, setting a higher burden for the derived model. To prove that the divide has not migrated with time, and justify the claims made currently, you need to show in this work that the radar data are *incompatible* with a solution in which the divide migrates. Is it possible to reproduce the isochron field with a temporally variable p' value?

The novel contribution of the second component of this work, discussing the role of surface slope and surface curvature on local surface mass balance variability, is not clearly articulated. The role of surface topography on snow trapping has been established for decades (Whillans 1975). It seems as though the basal influence on surface topography is the purview of a different study, leaving little for this study to discuss. A more quantitative treatment, or re-evaluation of our process understanding, is required to justify this section.

Finally, there is room for improvement in the clarity of the writing. While many paragraphs are well written, there are also many points throughout the paper where the logic or structure was unclear, making it hard to follow the flow of ideas to the authors conclusions. Individual comments on the writing are provided in the technical corrections.

**Technical Corrections:**

timeframe of its stability?

| Line #
12 | Comment
"site of the oldest as-yet-retrieved continuous ice core"                                                                                                                                                                                                                                                                                                                          |
|---------------------|-------------------------------------------------------------------------------------------------------------------------------------------------------------------------------------------------------------------------------------------------------------------------------------------------------------------------------------------------------------------------------------------------------------|
| 13-22               | The second half of this paragraph is too informal and largely unnecessary. For example, you do not need to clarify for the reader that the Dome has both an upward sloping flank and a downward sloping flank (that is the definition of a Dome).                                                                                                                                                           |
|                     | The point you are trying to make is that "total annual precipitation at Dome C is controlled
by the surface gradient and dominated by large precipitation events", but wading through
the superfluous information makes it hard to get to that point. This is a common complaint
I have with this manuscript.                                                                                      |
|                     | e.g.: "Modern precipitation at the ice core site is low (~XXmm/a), with infrequent storm events representing more than 50% of the total accumulation signal. Coastal air masses lose moisture as they are driven inland to higher elevation, resulting in a characteristic accumulation gradient with higher measured precipitation on the south side of Dome C."                                           |
| 23-37               | This paragraph starts with a discussion of dust provenance and ends with a sentence on the role of surface gradients in snow re-deposition. What is the point of this paragraph? Here and elsewhere in the paper, the writing seems unfocused. Decide what the point of that paragraph is supposed to be, and focus on that one idea.                                                                       |
| 41-43               | Medley et al. 2013 are reconstructing paleo-accumulation in only the highest part of the ice column, which is isolated from the strain-history and advection effects that make this particular study challenging. This section should be where you cite and discuss the large body of literature that focuses on deep-ice paleo-accumulation inversion, which is currently missing from the paper entirely. |
| 45                  | " show a continuous existence through historical timescales" This has an ambiguous antecedent (what is shown to exist? even though I know you mean existence of an accumulation gradient, you need to actually refer to the accumulation gradient here), and the word "historical" in "through historical timescales" is imprecisely defined. What is the                                                   |

58-63 Here again, the logic of this point is not clear. The last sentence, "several recent studies have shown the influence of increasing precipitation...", does not at all speak to the original statement, that knowing the position of a dome is crucial. In this list, it would make more sense to simply say:

"The position of topographic domes affects the spatial distribution of accumulation, which ultimately affects the geometry of the ice sheet (with its resulting sea-level implications) through time."

- 67 Citation needed here to justify the need to know flowlines of ice particles through time to interpret ice cores.
- 69-70 Remove this sentence, it doesn't speak to why knowing the position of the dome is crucial.
- 71-75 This point speaks directly to my biggest question about this work, but seems to be ignored through the rest of the paper. It will be hard for you to prove the dome position for the last 128 ka with this manuscript's approach, because there is no unique solution for divide position through time and accumulation rate through time. This problem is underconstrained.
- 85/87 Calling the topography and saddle "gentle" doesn't provide information to the reader. Either define gentle or remove.
- 102 "... center frequency of 60 MHz; internal isochrones are therefore coherent..." Coherent in what way? Are you saying, because it is 60 MHz, the wavelength is long enough that any changes in range between seasons are unresolvable by phase measurements? Coherent has a very specific, technical definition. Also, I would use just "isochrones", instead of "internal isochrones", as there is no such thing as an external isochrone.
- 117/149 "Pseudo-steady-state means that all parameters in the model are considered steady except for R(t)..." this statement about your model seems to violate the requirement set in line 71, in which you say "The location of the dome [through time] is required to model isochrones interpreted from radar surveys". With your model, you have functionally assumed a constant divide position. This arises again at line 149, when you assume tau is modeled perfectly. If the divide migrates, tau is not modeled correctly, and you are mapping variability in divide position into accumulation history. This seems like a fundamental flaw in your method.
- 172-178 "our paleoaccumulations are valid at the ice divide and the dome where horizontal ice flow speeds are negligible." I'm not sure the literature agrees that your assumptions hold over your domain. (Neumann et al. 2008) show that the vertical thinning through the column varies dramatically as you move further than one ice thickness away from the divide. As the divide moves around with time, the vertical thinning function changes, and the horizontal advection term becomes more important. Your domain extends nearly 100km from the modern divide, it is likely true that a 3D model is required to do this correctly. You need to provide more discussion of the region over which your assumptions are valid (or conversely, at what point these assumptions aren't valid). This is required for line 175 also, at what depth would your assumption that tau is fitted correctly break down? Without clearer justification, I am not convinced your assumptions regarding tau (and p' in the initial model formulation) hold.

- 180 "... the assumption of constant ice thickness is fair..." define fair. How good is it? What is the magnitude of error this might introduce?
- 201 What is "this data set"?
- 232 "marked decreasing gradient" is the accumulation rate gradient decreasing? Or the accumulation rate decreasing?
- 234-235 "... there is no clearly visible accumulation gradient over time..." I'm not sure what this means. Do you mean the gradient in time or the gradient in space? As in, the accumulation rates don't change systematically with time, or the accumulation (spatial) gradients don't change systematically with time?
- 244 "... the area of high accumulation is preceded by an area of low accumulation ..." preceded describes a temporal, or otherwise linearly ordered relationship - you are describing a 2D spatial relationship. "Adjacent to" would be better here.
- 247 "Paleoaccumulation rates per isochrone-bounded layer show a similar pattern in accumulations…" I'm not quite sure what pattern you are referring to. Are you saying the spatial pattern of paleoaccumulation is similar between time periods? Text needs to be clarified.
- 277-278 "As a reminder, these accumulation maps therefore display values of detrended paleoaccumulation once the large-scale precipitation gradients from ECMWF ERA-40 has been removed." This sentence is unnecessary. If you want to redefine what you mean by "detrended" in the previous sentence that is fine, but you don't need two sentences that both say "we have plotted the detrended data".
- 305-307 You state here that the meteorologic system and location of Dome C is stable, but that doesn't necessarily follow from the preceding statement. This is where having a clear set of criteria established, describing for the reader what aspect of your results *prove* the divide position is stable, is necessary. Also, as stated above, qualify what it means to be "stable".
- 308 "... must have interacted with the *same* surface topography ..." That claim is too grand. Draw any transect from the coast to the interior and you will find decreasing accumulation rates. South will always be further from the coast than North, so observing this gradient does little to prove the specifics of Dome C's position.
- 334 "This could be due to ice flow increasing with distance from the dome." It isn't clear what you mean by this.
- 355 "... radar data used *do* not show ..." data is a plural noun
- 366-368 "If the surface slope and curvature do not change, we can suppose this implies the position of the divide and the dome must have remained relatively stable." Why is this true? Small scale variability in the surface (controlling curvature on a 3km scale) seems decoupled from the divide position. As you've pointed out earlier, and in the following paragraphs, the small scale surface variability maps well into subglacial topography (and possibly enhanced by subglacial melt). These effects are separate from those controlling divide position, which is an controlled by continental ice dynamics. I don't think you've proven that the stability of small scale features == the stability of large scale features.

- 413-414 "Both suggest that the current surface topography of the Dome C region had not changed significantly over the last glacial cycle." This is a significant claim, and there is an equally significant burden of proof to make it. I don't believe this model, or the analysis presented here, is capable of proving this statement. Again, unless you can show that the data is *incompatible* with scenarios that include divide migration, you should not publish that the divide has been stationary for 128 ka.
- Figure 1 Eliminate "radar-lines" title. It may also be less ambiguous to refer to "gray contours" instead of "gray lines" when discussing the Bamber et al surface elevations (there are lots of gray lines).
- Figure 4 "N-S accumulation rates decreasing with distance from the Indian Ocean coast..."
- Figure 5 Define "very good agreement". "Black lines [not white lines] outline the same areas of small-scale high accumulation..."
- Figure 6 It took me a long time to make heads or tails of this figure. At the very least, the radar lines should be thicker, as it is hard to see the accumulation rates relative to the curvature. It would also help to emphasize specific parts of this figure, so the reader knows how to focus their attention.

**Review References:**

- Fahnestock, M. et al., 2001. Internal layer tracing and age-depth-accumulation relationships for the northern Greenland ice sheet. *Journal of Geophysical Research*, 106(D24), p.33,789-33,797.
- Koutnik, M.R. et al., 2016. Holocene accumulation and ice flow near the West Antarctic Ice Sheet Divide ice core site. *Journal of Geophysical Research: Earth Surface*, 121, pp.1–18.
- Neumann, T. a. et al., 2008. Holocene accumulation and ice sheet dynamics in central West Antarctica. *Journal of Geophysical Research*, 113(F2), pp.1–9.
- Parrenin, F. et al., 2017. Is there 1.5 million-year old ice near Dome C, Antarctica? *The Cryosphere Discussions*, (May), pp.1–16.

Whillans, I.M., 1975. Effect of inversion winds on topographic detail and mass balance on inland ice sheets. *J. Glaciol*, 14(70), pp.85–90. Available at:

http://www.igsoc.org/journal.old/14/70/igs\_journal\_vol14\_issue070\_pg85-90.pdf.

---

## Referee Comment (RC2) · Anonymous Referee #2 · 22 Aug 2017

This paper uses a suite of internal layers detected by airborne ice-penetrating radar to derive paleo-accumulation rates and their distribution across an area around Dome Concordia station, Antarctica. The paper relies on a companion paper by Parrenin et al. for a one-dimensional model which converts measured internal layer depth and geometry to an estimate of paleo accumulation rate across the region. The paper also compares the derived paelo-accumulation rates with the modern rates from the ECMWF ERA40 model.

My largest concern with this paper are the assumptions that a 1-D model provides an appropriate approximation to the vertical strain rate everywhere in the model domain and that advection from adjacent grid cells can be safely ignored. The manuscript does not go into the impacts of these assumptions in any significant detail, and I think

a quantitative analysis of that assumption is warranted.

There are a number of other studies that have aimed to recover non-steady and non-uniform accumulation rate histories from variations of the depth of internal layers, such as Waddington et al. (2007), Parrenin et al. (2006), Neumann et al. (2008), and Koutnik et al. (2016). I note the citation for the Parrenin paper in the manuscript, and include the other citations below.

An approximate advection criteria could be derived from the ratio of the along-track discretization distance of the data (1km; per line 117), the flow velocity (significantly variable spatially), and the length scale across which substantial changes in accumulation rate are expected. If the spatial accumulation rate were uniform in space, the proposed approach would probably work well. However, the authors note that there are substantial accumulation rate differences in the modern field, leading me to question over what length and time scales advection can be neglected. For example, if the flow velocity is on the order of 0.1 m/ yr, ice traverses a 1 km grid cell in 10 ka. This suggest that for time periods longer than some fraction of 10ka, the depth to internal layers in a particular 1km cell is impacted by accumulation rates in the cell(s) upstream. As the flow velocity is increased, the situation is exacerbated.

In addition, it's not clear to me that the model generates a credible vertical velocity profile away from the ice divide. I'd expect that velocity profile and strain rate to vary substantially within a few times the ice thickness. A similar quantitative analysis should be done to support (or update) this assumption.

I see this as the major limiting factor in this manuscript. Either the scope of the inversion could be restricted to those areas for which a 1-D model is appropriate over the time period investigated here (I'd estimate this region to be within a ice thickness of the current divide position, given a flow velocity of 0.01 m/yr and a grid cell of 1km), OR a 2-d inversion could be done building off the literature. As this analysis would directly impact the results of the current study, I am reluctant to endorse the the resulting

accumulation rate patterns or history or comparisons with the current accumulation rate field.

Other comments are:

1. The model used in this paper is currently also under review, making it difficult to evaluate the application of the model to the problems posed in this manuscript. In particular, I am concerned about the accumulation rate uncertainties (line 271) and what aspects are and are not included in that analysis.

2.The topic is certainly worthy of study and is scientifically interesting. The stability of ice domes and divides is of primary importance in reconstructing the ice sheet history as well as ice chore chronologies. I'd suggest also reading Marshall and Cuffey (2000) as being relevant to the work presented here.

3. The work using the EMCWF ERA40 model is interesting, and will be relevant to the resulting paleo-accumulation rates. This section is explained clearly, and I don't have substantial comments.

Waddington, E.D., T.A. Neumann, M.R. Koutnik, H.P. Marshall and D.L. Morse. 2007. Inference of accumulation-rate pattern from deep radar layers. Journal of Glaciology, 53(183), 694-712.

Neumann, T. A., H. Conway, S. F. Price, E. D. Waddington, G. A. Catania, and D. L. Morse. 2008. Holocene accumulation and ice sheet dynamics in central West Antarctica, Journal of Geophysical Research, 113, F02018, doi:10.1029/2007JF000764.

Koutnik, M.R., T.J. Fudge, H. Conway, E.D. Waddington, T.A. Neumann, K.M. Cuffey, C. Buizert, K.C. Taylor. 2016. Holocene accumulation and ice flow near the West Antarctic Ice Sheet Divide ice core site. Journal of Geophysical Research: Earth Surface, 121, pp.1–18.

Marshall, S.J. and K.M. Cuffey. 2000. Peregrinations of the Greenland ice sheet divide in the last glacial cycle: implications for central Greenland ice cores. EPSL 79(1),

73-90.

---

## Editor Comment (EC1) · K. Matsuoka (Editor) · 24 Aug 2017

Dear authors

Two reviewers provided detail comments to your manuscript. They both concern applicability of 1D ice-flow assumption. The companion paper, Parrenin et al., also mentions that 1D flow assumption is unlikely valid away from the divide and shows a reservation to take the model results as is for such regions. Therefore, I suggest authors to provide a full response to this point in particular.

Kenny Matsuoka TC/TCD Editor
* * *

---

## Author Comment (AC1) · 30 Nov 2017

**Response to reviewer comments on "Stable accumulation patterns around Dome C, East Antarctica, over the last glacial cycle":**

We would like to first thank the Editor and both anonymous reviewers for their very helpful reviews on the paper. We hope that we have satisfactorily responded to all comments, which can be found here below. We apologize in advance for any repetition in our answers below: we wanted to respond to comments point-by-point.

Note that:
(1) We have significantly shuffled the order of the paragraphs in the Discussion Section, for clarity.
(2) We have changed some results and discussion points to reflect that we are only looking at the last 73 ka of paleoaccumulation data limited to 5 km of horizontal movement. Some of the observations had to be removed as they are simply not included in our dataset anymore.
(3) We have therefore modified the title slightly to reflect this switch in the focus of the paper.
(4) We now detail the 0-10 ka interval in Fig.6, and show the older layers in Supplement 4.
(5) Brice Van Liefferinge has been added as a co-author for his help in calculating and setting the horizontal advection threshold, and significantly contributing to the manuscript.
(6) Equations (2) and (3) have been modified for clarity.

**Review #1:**

**General Comments:**

In this paper, the authors use the spatial pattern of model derived paleo-accumulation rates [generated as part of a companion study (Parrenin et al. 2017)] to constrain the processes controlling surface mass balance around Dome C, East Antarctica, and infer the long-term stability of the ice divide presently located there. A 1D model inversion computes three free parameters – the average accumulation rate, the average geothermal flux, and the average velocity shape function – given temporal variability in accumulation set by the time-series derived from the EPICA Dome-C core. Using the geothermal flux and velocity shape function from that inversion, the authors then recompute the time-history for the accumulation rate to perfectly match the observed isochrons. From this accumulation rate product, the authors make the claim that the "surface topography of the Dome C regions has not changed significantly over the last glacial cycle [128 ka]."

There is a history of researchers using modeled isochron position to determine paleoaccumulation rates near several of the deep ice cores in Antarctica and

Greenland, most notably at WAIS Divide (e.g., Koutnik et al. 2016; Neumann et al. 2008) and NGRIP (e.g., Fahnestock et al. 2001). Both this study and the companion paper are missing all reference to this body of work, which, if read, would have highlighted challenges associated with deriving a unique solution for accumulation rates that go undiscussed in the current manuscript. Most notably, I think the authors of the previous works would take issue with the following model choice, stated in the companion paper:

"We use a 1D pseudo-steady (Parrenin et al., 2006) ice flow model, which assumes that the geometry, the shape of the vertical velocity profile, the ratio $\mu=m/a$ and the relative density profile are constant in time." (Line 104-105, (Parrenin et al. 2017))

As seen from the work at WAIS divide (Koutnik et al. 2016), the accumulation history trades-off non-uniquely with the local horizontal stretching rate and burial history (which is dependent on distance from divide), so by holding their "p" shape value constant in time and ignoring horizontal advection, the authors of this work prescribe a result with no changes in flow regime (ie, unchanging distance to the divide) at every position. This makes it impossible to constrain the actual stability with their results as presented, as their conclusion that the divide is stable is baked into their model assumptions.

In the absence of the ability to prove that the Dome C divide has been unchanging for 128 ka, I find it difficult to see the novel contribution of this paper.

**Specific Comments:**

The primary limitation of this paper is in the model formulation (which itself is under-review, making this manuscript harder to evaluate in isolation). Previous literature has documented a more thorough approach for this research question. The analysis of isochrons connected to the WAIS divide ice core included an exhaustive exploration of model space, varying both divide position and flow dynamics. Even over a 9.2 ka reconstruction, non-uniqueness in the WAIS divide solution made it difficult to make claims about accumulation and divide position with high certainty. Constraining results over 128 ka will be proportionally more difficult.

The 1D model has now been published which should answer part of these comments.

To reach the level of rigor demonstrated in previous studies, this work needs to do the following:

1) **Define "stable" –**

This manuscript describes the accumulation pattern and divide position as "relatively stable" several times, without any context for what that term means. Migrates or evolves at rates less than X? Has not moved beyond a range greater than X over time period Y? There is a general lack of precision in both the discussion and conclusions that must be refined.

We have now removed conclusions on divide position (see specific comments for details).
When we describe a "relatively stable" accumulation rate pattern, we mean that the pattern remains spatially stationary, i.e. moves horizontally on a scale that is significantly smaller than the length scale of the accumulation patterns observed (large-scale and small-scale).
However, since we have changed the scope of the paper to cover only the last 73 kyrs to minimize issues with horizontal advection, a lot of the vocabulary used has been changed.

**2) Clearly establish criteria for evaluating stability –**

In the current draft, the authors have not done enough to identify how they have inferred divide stability. My reading of the draft shows them making two claims: (1) spatial patterns in small-scale accumulation variability have been persistent through time, and (2) large scale accumulation gradients consistently show lower accumulation to the South. Without getting into the accuracy of the model produced accumulation rates, it is not clear that either of these conclusions justify an inference of divide stability. The authors ascribe small scale surface variability (which controls local mass balance) to processes at the bed, which means their temporal stability is not at all controlled by divide position. The large-scale accumulation pattern seen today depends mostly on relative distance to the coast. Without a clear marker for divide position in the current accumulation field, along with a demonstration that such a transition has not migrated with time, information in the large-scale gradient also seems insufficient to prove divide position has not changed.

We agree with the reviewer that we were too quick on affirming divide stability. We have now removed this affirmation in several parts of the manuscript (see specific comments section for details). We now only refer to the spatially stationarity/stability of the spatial pattern of the accumulation rate.

**3) Evaluate divide stability in the context of other evidence –**

A perfectly stable divide over the last 128 ka would develop a prominent, measurable Raymond arch. Given no Raymond arch presented, my suspicion is that there is not one under the current divide, which should provide evidence for at least

some instability in the system. Given this observation, the authors could provide a bound for *the most* stable the divide could be.

Following our answer to point (2) of the reviewer, evaluating the degree of instability of the divide's position is beyond this study.

**4) Explore the possibility of a temporally variable divide position with your model –**

This represents one of the biggest hurdles for the authors going forward. As written, this inverse model formulation is not suitably justified. This is in part because the model development is established in the companion paper, but the objectives of that paper and this one are quite different. In (Parrenin et al. 2017), the goal is to constrain the age of the deep ice. For that work, trade-offs in the strain thinning and accumulation rates do not matter, as their *combined* effect dictates the age of the ice. For the inferences you try and make in this paper, you need a unique solution that disentangles ice-flow effects from accumulation, setting a higher burden for the derived model. To prove that the divide has not migrated with time, and justify the claims made currently, you need to show in this work that the radar data are *incompatible* with a solution in which the divide migrates. Is it possible to reproduce the isochron field with a temporally variable p' value?

Again, we removed our conclusions regarding divide stability from the manuscript. We now concentrate on the stability of the accumulation pattern (both small-scale and large-scale).
Regarding how we disentangle the effects of accumulation and thinning, we reckon that for the first layer whose average depth is ~150 m, that is ~5 % of the total depth, the error in the thinning is small enough to not pollute significantly our accumulation results. Total thinning for this layer is never below 0.9. For the other layers, it is difficult to imagine an error in the thinning function that would produce, by chance, a similar pattern to that of the first layer.

We state this explicitly in the text in our Discussion:

*In the 1D pseudo-steady ice flow model described in the companion paper (Parrenin et al., 2017), the goal is to constrain the age of the deep ice. For that work, trade-offs in the strain thinning (i.e. p and $G_0$) and accumulation rates do not matter, as their combined effects dictate the age of the ice. However, to calculate the layer paleoaccumulation rates, we have to assume that tau (Eq.4) is fitted perfectly, which breaks down as horizontal advection increases. We reckon that for the first layer whose average depth is ~150 m, that is ~5% of the total depth, the error in the thinning is small enough to not pollute significantly our accumulation results (total thinning is always above 0.9). For the other*

*layers, it is difficult to imagine an error in the thinning function that would produce, by chance, a similar pattern to that of the first layer.*

The novel contribution of the second component of this work, discussing the role of surface slope and surface curvature on local surface mass balance variability, is not clearly articulated. The role of surface topography on snow trapping has been established for decades (Whillans ,1975). It seems as though the basal influence on surface topography is the purview of a different study, leaving little for this study to discuss. A more quantitative treatment, or re-evaluation of our process understanding, is required to justify this section.

We think it's relevant to discuss the relationship between the surface slope and curvature (in the direction of the prevailing wind) and the accumulation rates in this manuscript. There is a good correlation between sites of high positive curvature and high accumulation, as seen in Fig.6, and we compare our observations to published literature in the Discussion section (as cited in the manuscript). We agree with the reviewer that is it beyond the scope of this paper to go any deeper into snow trapping processes, but we think it's relevant to highlight the role of surface curvature on the accumulation patterns for future work. We add a few word to this effect in the Discussion:

*Although we cannot yet explain the mechanisms causing the small-scale paleoaccumulation variability we observe in the Dome C region, **which is beyond the scope of this manuscript**, our observations have important ramifications for better understanding the region's stability through time.*

Finally, there is room for improvement in the clarity of the writing. While many paragraphs are well written, there are also many points throughout the paper where the logic or structure was unclear, making it hard to follow the flow of ideas to the authors conclusions. Individual comments on the writing are provided in the technical corrections.

We hope to have satisfactorily streamlined the ideas in the manuscript. See specific responses here below.

**Technical Corrections:**

| Line # | Comment |
| --- | --- |
| 12 | "site of the oldest as-yet-retrieved *continuous* ice core" |

This has been added.

| 13-22 | The second half of this paragraph is too informal and largely unnecessary. For example, you do not need to clarify for the reader |

that the Dome has both an upward sloping flank and a downward sloping flank (that is the definition of a Dome).

The point you are trying to make is that "total annual precipitation at Dome C is controlled by the surface gradient and dominated by large precipitation events", but wading through the superfluous information makes it hard to get to that point. This is a common complaint I have with this manuscript.

e.g.: "Modern precipitation at the ice core site is low (~XXmm/a), with infrequent storm events representing more than 50% of the total accumulation signal. Coastal air masses lose moisture as they are driven inland to higher elevation, resulting in a characteristic accumulation gradient with higher measured precipitation on the south side of Dome C."

This has been modified to be more concise as suggested:

*Modern surface precipitation on the Dome C plateau is extremely low (~25 mm $yr^{-1}$ Stenni et al., 2016), with infrequent storm events representing more than 50% of the total annual precipitation (Frezzotti et al., 2005). Coastal air masses lose moisture as they are driven inland to higher elevation, resulting in a characteristic precipitation gradient with higher measured and modeled precipitation on the north side of Dome C (Arthern et al., 2006; Genthon et al., 2016; Kållberg et al., 2004, Gallée et al., 2013; Palerme et al., 2014; Van Wessem et al., 2014).*

23-37   This paragraph starts with a discussion of dust provenance and ends with a sentence on the role of surface gradients in snow re-deposition. What is the point of this paragraph? Here and elsewhere in the paper, the writing seems unfocused. Decide what the point of that paragraph is supposed to be, and focus on that one idea.

We have now modified the paragraph to focus on precipitation redistribution by wind and have added the Whilland (1975) citation which was interesting and followed up on Black and Budd's work:

*Snow precipitation is homogeneous at a large-scale, whereas local variations in snow accumulation are controlled by local surface topography as a function of wind direction. Black and Budd (1964) and Budd (1971) are the first to observe the close relationship between bedrock relief, surface slope and accumulation rates in Wilkes Land. Whillans (1975) details how wind speed and direction can affect total mass balance in Marie Byrd Land. Frezzotti et al. (2007) show that*

*surface slope in the prevailing wind direction (SPWD) is a key constraint in determining spatial and temporal variability of precipitation; a higher SPWD can lead to significant ablation and redeposition of snow (Frezzotti et al., 2002b, 2002a, 2005, 2007). Das et al. (2013) show that SPWD is a strong threshold for the formation of wind scour or megadune fields. Evidence for a persistent westerly wind circulation pattern comes from mineral dust measured at EPICA Dome C which show a uniform geographic provenance from South America and Australia to the East Antarctic plateau during glacial-interglacial cycles (Delmonte et al., 2010; Albani et al., 2012).*

41-43          Medley et al. 2013 are reconstructing paleo-accumulation in only the highest part of the ice column, which is isolated from the strain-history and advection effects that make this particular study challenging. This section should be where you cite and discuss the large body of literature that focuses on deep-ice paleo-accumulation inversion, which is currently missing from the paper entirely.

We completely agree that this literature should have been included in this manuscript. This body of work is now referred to explicitly in the paragraph:

*Because the internal stratigraphy represents isochronal surfaces throughout much of the ice sheet, dated internal radar reflectors can be used to directly constrain the surface mass balance of the ice sheet* **in the highest part of the ice column (Medley et al., 2013). Reconstructing accumulation history from deeper isochrones is more ill-posed as both accumulation variations and changes in ice flow can affect isochrone geometries (e.g. Koutnik et al., 2016; Neumann et al., 2008; Parrenin and Hindmarsh, 2007; Parrenin et al., 2006; Waddington, et al., 2007; Nereson and Waddington, 2002), and certain assumptions have to be made about one or the other (Martin et al., 2009; Leysinger Vieli et al., 2011; Morse et al., 1998, see companion paper for more discussion). Assumptions on the vertical strain rate will also affect reconstructed paleoaccumulation rates (e.g. Macgregor et al., 2015; Waddington et al., 2007).**

The companion paper already includes a lot of the discussion on deep-ice paleoaccumulation inversions, therefore we keep our discussion brief.

Furthermore, to avoid increasingly ill-posed conditions with deep-ice paleoaccumulation inversions, we now only show paleoaccumulation rates where the magnitude of the horizontal advection for the time span represented by the layer is insignificant with respect to the scale of the observed accumulation variations.
In other words: since the small scale areas of high accumulation described are ~20 km wide, we allow a maximum of 5 km of horizontal advection for the layer's entire climate history. This threshold of 5 km is imposed to remove any paleoaccumulation

data point that has experienced more than 5 km of horizontal advection, for each isochrone-bounded layer.

We use published ice surface balance velocities (Van Liefferinge & Pattyn, 2013) and the layers' mean age and depth to create paleoaccumulation masks for each layer. Since the ice surface balance velocities represent present-day conditions, we scale them by the AICC2012 accumulation variations during each layer's mean age with respect to present. Ice surface velocities are therefore proportionally smaller during the glacial cycle.

Of course, the 1D assumption made to calculate the paleoaccumulation rates is not perfect. But we argue that by imposing these data constraints, our manuscript represents the best attempt at using a 1D model to constrain paleoaccumulation rates in the region. The development of a 3D model with further data constraints in the region is our aim, but we argue that the observations made in this manuscript are a step forward in understanding the Dome C region.

We describe this in the methods section which has be reordered slightly (the paragraph on uncertainties is now moved to after this discussion):

*To respect our assumption that # is modeled perfectly, we only calculate paleoaccumulation rates $abar_{0,\Delta x}$ for the first four isochrone-bounded layers. Our fourth and deepest layer used reaches an average depth of 30% of the ice thickness, with calculated thinning never reaching below 0.6. Furthermore, to avoid ill-posed conditions for our 1D paleoaccumulation reconstructions, we only retain data points that have experienced a maximum of 5 km of horizontal advection. We do this for each 5 155 layer, using Van Liefferinge and Pattyn (2013) ice surface balance velocities, corrected for temporal velocity variations using R(t) (Parrenin et al., 2017), and the age interval spanned by the layer considered. Any point that has traveled more than 5 km horizontally is masked.* [...]
*Care must be taken in not over-interpreting the paleoaccumulation maps obtained. We do not argue that we have reconstructed absolute paleoaccumulations for the past 73 kyrs. The 1D pseudo-steady ice flow model used here (see companion paper, Parrenin et al., 2017) does not take horizontal advection into account. Paleoaccumulation rates calculated are valid at the ice divide and the dome where horizontal ice flow speeds are negligible. Farther away, horizontal advection has a larger influence. A full 3D model is required to reconstruct accumulation rates more extensively in space and further in time.*

We also describe this in the Discussion section:

*The 1D assumption to calculate paleoaccumulation rates is clearly the largest source of uncertainty in our reconstructions. In the 1D pseudo-steady ice flow*

*model described in the companion paper (Parrenin et al., 2017), the goal is to constrain the age of the deep ice. For that work, trade-offs in the strain thinning (i.e. p and G0) and accumulation rates do not matter, as their combined effects dictate the age of the ice. However, to calculate the layer-by-layer paleoaccumulation rates, we have to assume that (Eq.4) is fitted perfectly, which breaks down as horizontal advection increases. We reckon that for the first layer whose average depth is ~150 m, that is ~5% of the total depth, the error in the thinning is small enough to not pollute significantly our accumulation results (total thinning is always above 0.9). For the other layers, it is difficult to imagine an error in the thinning function that would produce, by chance, a similar accumulation pattern to that of the first layer. In addition, by setting a limit on the maximum horizontal advection allowed for each age interval, the described accumulation patterns and variations are reasonably unaffected by the 1D assumption. The threshold of 5 km is chosen such that horizontal advection is negligible compared to the scale of the observed accumulation rate variability. The small-scale areas of high accumulation are at least 20 km wide in the region, therefore the 5 km threshold on horizontal movement does not affect our conclusions. We are only able to reconstruct paleoaccumulation rates back through 73 ka, therefore a 3D model is required to look at paleoaccumulation rates further back in time.*

*Furthermore, the model assumes a constant ice thickness through time. Even though small variations in the ice thickness through time will affect the absolute value of the reconstructed accumulation rates, the assumption of constant ice thickness is fair for the center of the EAIS where modeled ice thickness variations have been reported below 200 m (Bentley, 1999; Ritz et al., 2001; Parrenin et al., 2007)* **and little is known of the spatial distribution of these ice thickness variations in the center of the ice sheet. A 5% error on the ice thickness will produce a 5% error on the thinning function (Parrenin et al., 2007) and therefore a 5% error on the accumulation rates calculated. This error can be ignored for two reasons. First, it is small compared to the accumulation variations that we observe (larger than 10%). And second, it only affects the absolute value of the accumulation rates reconstructed but not the relative differences in accumulation rates from one location to the next in the Dome C region. Since we focus exclusively on changes in gradients and patterns in accumulation rates, this additional source of error doesn't affect our conclusions. Despite this error, we observe a clear reduction in the magnitude of the accu mulation rates as we go back in time and enter the last glacial maximum, as expected and measured in ice cores (Bazin et al., 2013; Veres et al., 2013; Parrenin et al., 2017).**

45          "… show a continuous existence through historical timescales…" This has an ambiguous antecedent (what is shown to exist? even though I

know you mean existence of an accumulation gradient, you need to actually refer to the accumulation gradient here), and the word "historical" in "through historical timescales" is imprecisely defined. What is the timeframe of its stability?

*This has been made clearer:*

*Verfaillie et al. (2012) show a continuous existence **of a precipitation gradient** through **the last 300 year**,*

58-63      Here again, the logic of this point is not clear. The last sentence, "several recent studies have shown the influence of increasing precipitation…", does not at all speak to the original statement, that knowing the position of a dome is crucial. In this list, it would make more sense to simply say:

     "The position of topographic domes affects the spatial distribution of accumulation, which ultimately affects the geometry of the ice sheet (with its resulting sea-level implications) through time."

*The last sentence has been removed, and the item has been changed as suggested:*

***The spatial distribution of snow accumulation affects the position of topographic domes, which ultimately affects the geometry of the ice sheet (with its resulting sea-level implications) through time (Scarchilli et al., 2011; Fujita et al., 2011; Morse et al., 1998).***

67      Citation needed here to justify the need to know flowlines of ice particles through time to interpret ice cores.

*Citations have been added:*

***Constraints on accumulation and flowline geometries of ice particles through time are necessary*** *to reconstruct ice core chronologies and correct for the effects associated with deposition at a different location and elevation than the ice coring site **(Koutnik et al., 2016; Parrenin et al., 2007).***

69-70      Remove this sentence, it doesn't speak to why knowing the position of the dome is crucial.

*We agree, it has been removed.*

71-75      This point speaks directly to my biggest question about this work, but seems to be ignored through the rest of the paper. It will be hard for

you to prove the dome position for the last 128 ka with this manuscript's approach, because there is no unique solution for divide position through time and accumulation rate through time. This problem is underconstrained.

We have removed the suggestion of a stable position of the dome in the following paragraph according to early comments:

*Here, we reconstruct paleoaccumulation rates for the Dome C region using a 1D pseudo-steady ice flow model (Parrenin et al., 2017, described in the companion paper) **for the last 73 kyrs** using the isochronal constraints obtained from radar surveys. We discuss the large-scale accumulation and small-scale variations in accumulation **calculated around Dome C**. We do not attempt to reconstruct older paleoaccumulations due to the 1D assumptions **and the increasing horizontal advection with depth**.*

85/87         Calling the topography and saddle "gentle" doesn't provide information to the reader. Either define gentle or remove.

We defined gentle previously but have moved the description around to make it clearer:

*The topography **of the Dome C region** is gentle: **the change in elevation is ~10 m across 50 km (Genthon et al., 2016), reaching a maximum elevation at Dome C of ~3266 m above sea level (geoid height)**.*

And we have also removed "gentle" from the following sentence, as it is clear at this point that the surface topography is more or less flat:

*A **saddle connects** Dome C to Lake Vostok along the ice divide, with a secondary dome referred to as "Little Dome C" (LDC) just south of the Dome C ice core site.*

102         "... center frequency of 60 MHz; internal isochrones are therefore coherent..." Coherent in what way? Are you saying, because it is 60 MHz, the wavelength is long enough that any changes in range between seasons are unresolvable by phase measurements? Coherent has a very specific, technical definition. Also, I would use just "isochrones", instead of "internal isochrones", as there is no such thing as an external isochrone.

By "coherent", we meant that all the data has the same vertical resolution therefore isochrones can be matched from one season to the next. The effect of annual accumulation at the surface on isochrone matching across radar surveys from different seasons is insignificant when considering the ~8 m vertical resolution of

the radar system, and the very low precipitation rates in the region (~25 mm yr$^{-1}$ as stated earlier in this response and the manuscript).
We have modified the manuscript in this way:

*All surveys use the same center frequency of 60 MHz,* **and the same bandwidth of 15 MHz; radar isochrones can therefore be easily matched from one season to the next.**

And we have also changed "internal isochrone" to "isochrone", and in all other places it appeared in the manuscript.

117/149      "Pseudo-steady-state means that all parameters in the model are considered steady except for R(t)…" this statement about your model seems to violate the requirement set in line 71, in which you say "The location of the dome [through time] is required to model isochrones interpreted from radar surveys". With your model, you have functionally assumed a constant divide position. This arises again at line 149, when you assume tau is modeled perfectly. If the divide migrates, tau is not modeled correctly, and you are mapping variability in divide position into accumulation history. This seems like a fundamental flaw in your method.

We realized that defining dome position from our 1D model and available constraints was an ill-posed problem, and perhaps jumped too quickly to conclusions on dome stability and so have removed all implications on dome stability. However, reconstructing time-averaged accumulation rates and paleoaccumulation rates is not ill-posed for the shallowest four layers as the horizontal distance travelled by these layers is small compared to the accumulation patterns described. As described further up, we now apply a limit of 5km of horizontal advection for any layer data points using balance velocities provided by Van Liefferinge & Pattyn, 2013. The assumptions made are now better detailed and discussed in the Discussion section.

172-178      "our paleoaccumulations are valid at the ice divide and the dome where horizontal ice flow speeds are negligible." I'm not sure the literature agrees that your assumptions hold over your domain. (Neumann et al. 2008) show that the vertical thinning through the column varies dramatically as you move further than one ice thickness away from the divide. As the divide moves around with time, the vertical thinning function changes, and the horizontal advection term becomes more important. Your domain extends nearly 100km from the modern divide, it is likely true that a 3D model is required to do this correctly. You need to provide more discussion of

the region over which your assumptions are valid (or conversely, at what point these assumptions aren't valid). This is required for line 175 also, at what depth would your assumption that tau is fitted correctly break down? Without clearer justification, I am not convinced your assumptions regarding tau (and p' in the initial model formulation) hold.

This is the same answer as further up: we removed our conclusions regarding divide stability from the manuscript. We now concentrate on the stability of the accumulation pattern (both small-scale and large-scale).
Regarding how we disentangle the effects of accumulation and thinning, we reckon that for the first layer whose average depth is ~150 m, that is ~5 % of the total depth, the error in the thinning is small enough to not pollute significantly our accumulation results. For the other layers, it is difficult to imagine an error in the thinning function that would produce, by chance, a similar pattern to that of the first layer. The deepest layer used to calculate paleoaccumulation rates reaches ~30% of the ice thickness, with a strain thinning value above 0.6 (see comments above).

We state this explicitly in the text in our Discussion:

*In the 1D pseudo-steady ice flow model described in the companion paper (Parrenin et al., 2017), the goal is to constrain the age of the deep ice. For that work, trade-offs in the strain thinning (i.e. p and G0) and accumulation rates do not matter, as their combined effects dictate the age of the ice. However, to calculate the layer-by-layer paleoaccumulation rates, we have to assume that # (Eq.4) is fitted perfectly, which breaks down as horizontal advection increases. We reckon that for the first layer whose average depth is ~150 m, that is ~5% of the total depth, the error in the thinning is small enough to not pollute significantly our accumulation results (total thinning is always above 0.9). For the other layers, it is difficult to imagine an error in the thinning function that would produce, by chance, a similar accumulation pattern to that of the first layer. In addition, by setting a limit on the maximum horizontal advection allowed for each age interval, the described accumulation patterns and variations are reasonably unaffected by the 1D assumption. The threshold of 5 km is chosen such that horizontal advection is negligible compared to the scale of the observed accumulation rate variability. The small-scale areas of high accumulation are at least 20 km wide in the region, therefore the 5 km threshold on horizontal movement does not affect our conclusions. We are only able to reconstruct paleoaccumulation rates back through 73 ka, therefore a 3D model is required to look at paleoaccumulation rates further back in time.*

Furthermore, we want to mention that current models are not necessarily robust enough to accurately define the position of the dome and divide through time. E.g. Pollard et al. produced a static Dome C and divide over the last 20 kyrs (~40 km,

which is within the grid cell resolution) in their 2009 model, while their 2015 model update shows a ~200 km migration of the divide. In a more recent model (FETISH, Pattyn et al., 2017), preliminary test reconstructions over the last 40 kyrs do not show such variability (*pers. comm.* Brice Van Liefferinge). These discrepancies have never been discussed or reconciled.

In this manuscript, we therefore choose to assume a relatively stable ice flow configuration for the Dome C region until the contrary is demonstrated.

180 "… the assumption of constant ice thickness is fair…" – define fair. How good is it? What is the magnitude of error this might introduce?

We argue that this assumption is "fair" because we only detail gradients and patterns in accumulation rates through the region. If the ice thickness of the ice sheet has varied ~200 m through time, this represents a 5% change in total ice thickness change. This will produce also a 5% error on the thinning function (Parrenin et al., 2007), which means we have a 5% error on the accumulation rates. However, this error is small compared to the accumulation variations that we observe (which are larger than 10%). This error will affect the absolute value of the accumulation rates reconstructed but not the relative differences in accumulation rates from one location to the next in the Dome C region, and since we are interested in changes in **gradients and patterns** in accumulation rates, this additional source of error shouldn't affect our conclusions. We do not include this uncertainty in calculating our uncertainty on the steady-state accumulation rate for that reason but we discuss it in our Supplement 2:

*Additional error arising from assuming a constant ice thickness is not taken into account in these uncertainties. However, it does not affect the spatial distribution of accumulation uncertainties, but rather increases the magnitude of accumulation uncertainties uniformly for the entire Dome C region. If we include this additional 5% error on accumulation rates (see manuscript): the area east of the CR (where uncertainties are already highest) has a resulting rms error of 1.2 mm-we yr$^{-1}$. Compared to the small-scale accumulation rate variations observed which represent accumulation differences of ~5 mm-we yr$^{-1}$ (see Fig.3 of the manuscript), this source of error is negligible*

Knowledge of the ice sheet thickness' evolution in this region is poor as "direct" measurements (exposure ages) can only be made along the edges of the ice sheet, where nunataks pierce the ice sheet cover (Bentley et al., 1999; Whitehouse et al., 2012; …). Ice thickness variations in the center of the ice sheet have only been modeled with a coarse horizontal resolutions (e.g. Whitehouse et al., 2012). For the Dome C region described here, the EDC ice core provides the only (modeled) constraints on ice thickness evolution (Parrenin et al., 2007). Ice thickness

reconstructions assume homogeneous variations on the scale of the region studied in this manuscript.

This discussion has been moved to the Discussion section and expanded to describe in a little more details why we choose to ignore ice thickness variations:

*Furthermore, the model assumes a constant ice thickness through time. Even though small variations in the ice thickness through time will affect the absolute value of the reconstructed accumulation rates, the assumption of constant ice thickness is fair for the center of the EAIS where modeled ice thickness variations have been reported below 200 m (Bentley,1999; Ritz et al., 2001;* **Parrenin et al., 2007***) **and little is known of the spatial distribution of these ice thickness variations in the center of the ice sheet. A 5% error on the ice thickness will produce a 5% error on the thinning function tau (Parrenin et al., 2007) and therefore a 5% error on the accumulation rates calculated. This error can be ignored for two reasons. First, it is small compared to the accumulation variations that we observe (larger than 10%). And second, it only affects the absolute value of the accumulation rates reconstructed but not the relative differences in accumulation rates from one location to the next in the Dome C region. Since we focus exclusively on changes in gradients and patterns in accumulation rates, this additional source of error doesn't affect our conclusions. Despite this error, we observe a clear reduction in the magnitude of the accumulation rates as we go back in time and enter the last glacial maximum, as expected and measured in ice cores (Bazin et al., 2013; Veres et al., 2013; Parrenin et al., 2017).***

201        What is "this data set"?

This is now explicitly stated:

*A number of steps went into* **adjusting the ECMWF ERA40 modeled precipitation rates to field measurements, to calculate the "ECMWF ERA40 estimated present-day surface accumulation rates"**, *shown in Fig.4.* **These steps are:**

We have also adjusted the sentence lines 210-211:

*For this, we calculate a quadratic fit of the ECMWF ERA40* **present-day** *surface accumulation values* **(calculated as** *described above) with each isochrone-bounded layer's*

232        "marked decreasing gradient" – is the accumulation rate gradient decreasing? Or the accumulation rate decreasing?

This has been changed to accumulation rate:

*The A-A' radar line runs along the ice divide, and a marked decreasing **accumulation rate** can be seen going from the northeast side towards the southwest consistently over all age intervals.*

234-235    "... there is no clearly visible accumulation gradient over time…" I'm not sure what this means. Do you mean the gradient in time or the gradient in space? As in, the accumulation rates don't change systematically with time, or the accumulation (spatial) gradients don't change systematically with time?

We meant there is no visible spatial gradient in accumulation and it does not change with time. We now write:

*This transect runs across the divide and there is no clearly visible **spatial** accumulation gradient **for all age intervals, except a weaker one for the interglacial 10 ka isochrone. This is expected as the southern end of this radar line is on the high-accumulation side of the divide.***

244    "... the area of high accumulation is preceded by an area of low accumulation …" preceded describes a temporal, or otherwise linearly ordered relationship - you are describing a 2D spatial relationship. "Adjacent to" would be better here.

We have changed "preceded" to "adjacent to".

247    "Paleoaccumulation rates per isochrone-bounded layer show a similar pattern in accumulations…" I'm not quite sure what pattern you are referring to. Are you saying the spatial pattern of paleoaccumulation is similar between time periods? Text needs to be clarified.

We have now clarified the text:

***The spatial pattern of paleoaccumulation rates per isochrone-bounded layer (Fig.4) is similar to that of the time-averaged accumulation**: a large-scale gradient N-S with superimposed areas of higher accumulation in the same locations as for the time-averaged **accumulation** reconstruction. We note a striking similarity between the time-averaged accumulation rate (Fig.3) and the paleoaccumulation rates for the ages 0 ka – 10 ka (Fig.4). We also note that **accumulation rates are higher for the interglacial age interval (0 ka - 10 ka in Fig.4).***

277-278    "As a reminder, these accumulation maps therefore display values of detrended paleoaccumulation once the large-scale precipitation gradients from ECMWF ERA-40 has been removed." This sentence is unnecessary. If you want to redefine what you mean by "detrended" in

the previous sentence that is fine, but you don't need two sentences that both say "we have plotted the detrended data".

This sentence has now been removed as we refer the reader to the relevant sections further up.

305-307    You state here that the meteorologic system and location of Dome C is stable, but that doesn't necessarily follow from the preceding statement. This is where having a clear set of criteria established, describing for the reader what aspect of your results prove the divide position is stable, is necessary. Also, as stated above, qualify what it means to be "stable".

We agree that we over-reached our conclusions. We only refer to the stability of the moisture provenance based on the consistent large-scale accumulation gradient observed over all age intervals.

We now write:

*The fact that our paleoaccumulation reconstructions reproduce **the** present-day **large-scale** surface accumulation gradient and that this remains true **back to 73 ka suggests persistence of the source of moisture for this part of the East Antarctic plateau through the last glacial and deglaciation**.*

308    "… must have interacted with the *same* surface topography …" That claim is too grand. Draw any transect from the coast to the interior and you will find decreasing accumulation rates. South will always be further from the coast than North, so observing this gradient does little to prove the specifics of Dome C's position.

We agree that our statement was too grand in implying stable topography of the Dome C region. However, we argue that based on Fig. 2 and Fig. 4, there is a clear orientation of the accumulation gradient north south. Transect B-B' in Fig. 2 shows that there is little E-W contribution to the gradient. Based on work done by Scarchilli et al and Frezzotti et al, we propose that moisture source trajectories persisted over the last 73 kyrs. This is an important piece of information when comparing to paleo-atmospheric model outputs.

The paragraph has been modified accordingly and combined with the following paragraph to clarify the language:

*The observed patterns of paleoaccumulation agree well with previous studies of surface snow accumulation variability in the Dome C region.*

*Considering first the large-scale patterns in the accumulation reconstructions, we observe a consistent large-scale gradient (large-scale here refers to 100s of kilometers) for each age interval, with accumulation values decreasing from the north side of Dome C to the south side.* **(Scarchilli et al., 2011) suggest moisture provenance from the Indian Ocean sector is the most consistent with the clear north south gradient in precipitation observed as we near Dome C. The fact that our paleoaccumulation reconstructions reproduce the present-day large-scale surface accumulation gradient and that this remains true back to 73 ka suggests persistence of the source of moisture for this part of the East Antarctic plateau through the last glacial and deglaciation. Transects A-A' and B-B' in Fig.2 clearly show the north south orientation of the accumulation gradient. This large-scale accumulation gradient is also clearly seen in the ECMWF ERA40 data for the region (Fig.5), as well as in other large-scale accumulation models of the region (e.g. Genthon et al., 2016) or Regional Climate Model (MAR) (Gallee et al., 2013;Gallee et al., 2015). GPR data collected during traverses across Dome C and along the divide also show a strong north-south gradient in accumulation (Emmanuel Le Meur, pers. comm.; urbini et al., 2008; Verfaillie et al., 2012). We note a good agreement between our accumulation values and trends along A-A' going from Dome C along the ice divide towards Vostok (top panel of Fig.2) and the GPR transect measured by (Verfaillie et al., 2012) on the other side of the Dome C divide. A SPRI airborne transect collected over Dome C shows a strong accumulation gradient of 10s of mm yr-1 over a spatial scale of 100s of km (Siegert, 2003). Urbini et al., 2008) show a small component of counter-clockwise rotation of the accumulation pattern in historical times centered on Dome C, but the general north south gradient difference in accumulation across the dome remains. Measurements made in other areas of the ice sheet, e.g. across Talos Dome (Frezzotti et al., 2007), point to similar patterns: accumulation is highest near the moisture source and decreases with distance from the coast. (Fujita et al., 2011) point to the same patterns of reduced accumulation inland across Dronning Maud Land.**

334        "This could be due to ice flow increasing with distance from the dome." It isn't clear what you mean by this.

This was badly worded. We meant that as we consider deeper and deeper layers in the ice column, they've had more time to accumulate horizontal advection, which can go up to 5km based on the threshold we set (see further up). Therefore we do not expect a perfect spatial match of accumulation and present-day surface CPWD. This has been reworded:

*For any deeper layer (Fig.S4), this relationship is slightly offset in space; **a likely cause is the increased amount of horizontal advection with depth, up to the set maximum of 5 km.***

355          "… radar data used *do* not show …" - data is a plural noun

Changed.

366-368      "If the surface slope and curvature do not change, we can suppose this implies the position of the divide and the dome must have remained relatively stable." Why is this true? Small scale variability in the surface (controlling curvature on a 3km scale) seems decoupled from the divide position. As you've pointed out earlier, and in the following paragraphs, the small scale surface variability maps well into subglacial topography (and possibly enhanced by subglacial melt). These effects are separate from those controlling divide position, which is controlled by continental ice dynamics. I don't think you've proven that the stability of small scale features == the stability of large scale features.

We agree with the reviewer. As stated further up, we are also convinced that stating dome stability from our current model and data constraints is ill-posed. However, we suggest that a persistent small-scale accumulation pattern is a significant observation when we now apply the threshold of 5km of horizontal displacement in the time spanned by layer. The spatial stationarity of the small-scale areas of high accumulation means that there was a consistent interaction of surface topography with the moisture sources, sources whose trajectories had to be spatially unchanging through time, an important conclusion when modeling ice sheet evolution in the scope of retrieving a 1.5 million-year-old ice core.

We have modified this paragraph in the discussion to only explain future developments:

*Although we cannot yet explain the mechanisms causing the small-scale paleoaccumulation variability we observe in the Dome C region, **which is beyond the scope of this manuscript,** our observations have important ramifications **for better understanding the region's stability through time. In the future, we hope to improve our paleoaccumulation rate reconstructions, and in particular go back further into the last glacial cycle with a full 3D model.** Further GPR data was recently collected over the LDCm, and strain nets and various other instruments have been deployed. These new measurements will add to the existing data set and provide important constraints **if we hope to develop 3D inversions.***

And the points that you raise have been moved to the conclusion:

*We reconstructed accumulation rates for the last 73 kyrs. Looking at both large- and small-scale accumulation gradients, we show that these have not changed significantly over the last glacial cycle. Large-scale accumulation gradients will remain constant **if moisture-bearing air mass trajectory interactions with surface topography do not vary.** Small-scale accumulation variations are strongly controlled by SPWD and CPWD and therefore, if the pattern of high and low accumulations remains fixed over a long period of time, this requires **consistent interactions between local surface slopes and prevailing winds over the last 73 kyrs**, independent of whether the control comes from the bedrock topography and/or potential basal melting. This points to a spatially stationary and persistent accumulation pattern in the Dome C region over the last glacial, an important constraint for modeling efforts in the area, both for dating existing ice cores as well as for the prospecting of a 1.5 million-year-old ice core site.*

413-414    "Both suggest that the current surface topography of the Dome C region had not changed significantly over the last glacial cycle." This is a significant claim, and there is an equally significant burden of proof to make it. I don't believe this model, or the analysis presented here, is capable of proving this statement. Again, unless you can show that the data is *incompatible* with scenarios that include divide migration, you should not publish that the divide has been stationary for 128 ka.

We agree, and in line with comments above, have reformulated our conclusions to only point to accumulation rate persistence and not dome/divide stability. See comments above.

Figure 1    Eliminate "radar-lines" title. It may also be less ambiguous to refer to "gray contours" instead of "gray lines" when discussing the Bamber et al surface elevations (there are lots of gray lines).

The title has been removed, and "gray lines" has been changed to "gray contours".

Figure 4    "N-S accumulation *rates* decreasing with distance from the Indian Ocean coast…"

This has been changed.

Figure 5    Define "very good agreement". "Black lines [not white lines] outline the same areas of small-scale high accumulation…"

The wrong figure was uploaded. White lines are clearer for this figure so we now draw white lines.

Figure 6    It took me a long time to make heads or tails of this figure. At the very least, the radar lines should be thicker, as it is hard to see the accumulation rates relative to the curvature. It would also help to emphasize specific parts of this figure, so the reader knows how to focus their attention.

We have now changed the figure to only show the shallowest layer as it is the most extensive, and has also experienced the least amount of horizontal advection. We show the four layers as a supplement (S4) instead for those who are interested and mention in the results:

*We plot detrended paleoaccumulation for layers older than 10 ka, and observe that this relationship holds over the LDCM, with a slightly increasingly offset with increased ages (see S4).*

And in the Discussion section:

*For any deeper layer, this relationship is slightly offset in space; a likely cause is the increased amount of horizontal advection with depth, up to the set maximum of 5 km (see S4).*

**Review #2:**

This paper uses a suite of internal layers detected by airborne ice-penetrating radar to derive paleo-accumulation rates and their distribution across an area around Dome Concordia station, Antarctica. The paper relies on a companion paper by Parrenin et al. for a one-dimensional model which converts measured internal layer depth and geometry to an estimate of paleo accumulation rate across the region. The paper also compares the derived paleo-accumulation rates with the modern rates from the ECMWF ERA40 model.

My largest concern with this paper are the assumptions that a 1-D model provides an appropriate approximation to the vertical strain rate everywhere in the model domain and that advection from adjacent grid cells can be safely ignored. The manuscript does not go into the impacts of these assumptions in any significant detail, and I think a quantitative analysis of that assumption is warranted.

We agree that the assumption of the 1D model and that advection would be safely ignored was too optimistic. We have now restricted ourselves to data points that

have moved by 5 km horizontal or less. To do this, we used published ice surface balance velocities from Van Liefferinge & Pattyn (2013), modified using R(t) fluctuations from the companion paper to account for magnitude changes in the last glacial maximum, to set this 5 km limit, using the time interval spanned by each layer. This is better detailed in the manuscript in the method section:

*Care must be taken in not over-interpreting the paleoaccumulation maps obtained. We do not argue that we have reconstructed absolute paleoaccumulations for the past 73 kyrs. The 1D pseudo-steady ice flow model used here (see companion paper, Parrenin et al., 2017) does not take horizontal advection into account. Paleoaccumulation rates calculated are valid at the ice divide and the dome where horizontal ice flow speeds are negligible. Farther away, horizontal advection has a larger influence. A full 3D model is required to reconstruct accumulation rates more extensively in space and further in time.*

As well as in the Discussion section:

*The 1D assumption to calculate paleoaccumulation rates is clearly the largest source of uncertainty in our reconstructions. In the 1D pseudo-steady ice flow model described in the companion paper (Parrenin et al., 2017), the goal is to constrain the age of the deep ice. For that work, trade-offs in the strain thinning (i.e. p and G0) and accumulation rates do not matter, as their combined effects dictate the age of the ice. However, to calculate the layer-by-layer paleoaccumulation rates, we have to assume that tau (Eq.4) is fitted perfectly, which breaks down as horizontal advection increases. We reckon that for the first layer whose average depth is ~150 m, that is ~5% of the total depth, the error in the thinning is small enough to not pollute significantly our accumulation results (total thinning is always above 0.9). For the other layers, it is difficult to imagine an error in the thinning function that would produce, by chance, a similar accumulation pattern to that of the first layer. In addition, by setting a limit on the maximum horizontal advection allowed for each age interval, the described accumulation patterns and variations are reasonably unaffected by the 1D assumption. The threshold of 5 km is chosen such that horizontal advection is negligible compared to the scale of the observed accumulation rate variability. The small-scale areas of high accumulation are at least 20 km wide in the region, therefore the 5 km threshold on horizontal movement does not affect our conclusions. We are only able to reconstruct paleoaccumulation rates back through 73 ka, therefore a 3D model is required to look at paleoaccumulation rates further back in time.*

There are a number of other studies that have aimed to recover non-steady and non uniform accumulation rate histories from variations of the depth of internal layers, such as Waddington et al. (2007), Parrenin et al. (2006), Neumann et al. (2008), and

Koutnik et al. (2016). I note the citation for the Parrenin paper in the manuscript, and include the other citations below.

We respond to this in the comment above. We also now include these references as examples of studies that reconstruct non-uniform and non-steady accumulation history from deeper isochrones.

An approximate advection criteria could be derived from the ratio of the along-track discretization distance of the data (1km; per line 117), the flow velocity (significantly variable spatially), and the length scale across which substantial changes in accumulation rate are expected. If the spatial accumulation rate were uniform in space, the proposed approach would probably work well. However, the authors note that there are substantial accumulation rate differences in the modern field, leading me to question over what length and time scales advection can be neglected. For example, if the flow velocity is on the order of 0.1 m/ yr, ice traverses a 1 km grid cell in 10 ka. This suggest that for time periods longer than some fraction of 10ka, the depth to internal layers in a particular 1km cell is impacted by accumulation rates in the cell(s) upstream. As the flow velocity is increased, the situation is exacerbated.

We now only consider data points that have travelled a maximum distance of 5 km horizontally, using ice surface velocities and the amount of time the ice particles had to travel. For this, we use the mean age of each layer, and only keep data points that have travelled less than 5 km between its mean age and present day. Figures 4 , 6 and S4 therefore only display accumulation distributions back through to 73 ka. This point is answered in Review #1 in more detail and to avoid too much repetition, we refer the reviewer to those earlier answers.

In addition, it's not clear to me that the model generates a credible vertical velocity profile away from the ice divide. I'd expect that velocity profile and strain rate to vary substantially within a few times the ice thickness. A similar quantitative analysis should be done to support (or update) this assumption.

For the first isochrone whose average depth is ~150 m, that is 5 % of the total ice thickness, we reckon we can safely assume that the error in the thinning is negligible compared to the variations of surface accumulation rates. For the deeper layers, it might not be the case, but it is hard to imagine that we find a stability of the accumulation pattern by chance, due to errors in the calculated thinning functions. As described above, we now restrict ourselves to only using data points that have undergone 5 km of horizontal advection or less. We now describe this in the Methods section:

*To respect our assumption that tau is modeled perfectly, we only calculate paleoaccumulation rates bara$_{o,\Delta x}$* **for the first four isochrone-bounded layers. Our**

*fourth and deepest layer used reaches an average depth of 30% of the ice thickness, with modeled thinning rates never reaching below 0.6. Furthermore, to avoid ill-posed conditions for our 1D paleoaccumulation reconstructions, we only retain data points that have experienced a maximum of 5 km of horizontal advection. We do this for each layer, using Van Liefferinge & Pattyn (2013) ice surface balance velocities, corrected for temporal velocity variations using R(t) (Parrenin et al., 2017) and the age interval spanned by the layer considered. Any point that has traveled more than 5 km horizontally is masked.*

I see this as the major limiting factor in this manuscript. Either the scope of the inversion could be restricted to those areas for which a 1-D model is appropriate over the time period investigated here (I'd estimate this region to be within an ice thickness of the current divide position, given a flow velocity of 0.01 m/yr and a grid cell of 1km), OR a 2-d inversion could be done building off the literature. As this analysis would directly impact the results of the current study, I am reluctant to endorse the resulting accumulation rate patterns or history or comparisons with the current accumulation rate field.

See our detailed answer a few comments above on setting a 5 km threshold on horizontal advection for each layer.

Other comments are:
1. The model used in this paper is currently also under review, making it difficult to evaluate the application of the model to the problems posed in this manuscript. In particular, I am concerned about the accumulation rate uncertainties (line 271) and what aspects are and are not included in that analysis.

The model has now been accepted and published in TC, see https://doi.org/10.5194/tc-11-2427-2017.

2.The topic is certainly worthy of study and is scientifically interesting. The stability of ice domes and divides is of primary importance in reconstructing the ice sheet history as well as ice chore chronologies. I'd suggest also reading Marshall and Cuffey (2000) as being relevant to the work presented here.

Thank you for your recommendation. This paper was very interesting, it is what we had set out to do for the region. The hope is that future 3D work in the Dome C region would get us closer to answering the problematic of divide stability in the Dome C region.
We have been looking in more details at various model outputs describing the stability of the dome and divide in the region and found conflicting results.
E.g. Pollard et al. produces a static Dome C and divide over the last 20 kyrs (~40 km divide movement, which is within their grid's resolution) in their 2009 model, while

their 2015 model update shows a ~200 km migration of the divide. In a more recent model (FETISH, Pattyn et al., 2017), preliminary test reconstructions over the last 40 kyrs do not show such variability (*pers. comm.* Brice Van Liefferinge).
As these discrepancies have never been discussed or reconciled, we have to make the simplest assumption and choose to assume a relatively stable ice flow configuration for the Dome C region until the contrary is demonstrated.

3. The work using the EMCWF ERA40 model is interesting, and will be relevant to the resulting paleo-accumulation rates. This section is explained clearly, and I don't have substantial comments.

Thank you, we appreciate your positive remark.

**Editor comment:**

Two reviewers provided detail comments to your manuscript. They both concern applicability of 1D ice-flow assumption. The companion paper, Parrenin et al., also mentions that 1D flow assumption is unlikely valid away from the divide and shows a reservation to take the model results as is for such regions. Therefore, I suggest authors to provide a full response to this point in particular.

We have attempted to respond to the issues with the 1D assumption to the best of our ability. We have explained in details in our response to reviewers and in the paper why we estimate this assumption is valid in the region that we study. Especially after restricting ourselves to data points that have travelled 5 km or less. More data is needed to constrain dome stability, therefore we now only describe accumulation trends in the regions where the 1D assumption is most valid. We argue that by imposing these data constraints, our manuscript represents the best attempt at using a 1D model to constrain paleoaccumulation rates in the region. Further expansive present-day surface accumulation surface velocities are needed to answer the question of dome stability, as well as a 3D model, which we state specifically is our ultimate goal.

---

## Referee Report (RR1)

**Review Comments**

| Title: | Accumulation patterns around Dome C, East Antarctica, over the last glacial cycle |
|---|---|
| **Journal:** | The Cryosphere |
| **Date:** | 01/03/2018 |

**General Comments:**

This resubmission represents a significant improvement over the original work. The manuscript has been refocused, moving away from discussion of divide dynamics toward a more straightforward inference of accumulation rates and spatial patterns over the last 73 ka. This set of analyses is more appropriate for the model used, and the language in the manuscript provides the necessary caveats to prevent readers from over-interpreting the authors' results. These improvements are reflected in the detailed response to reviewers provided, which shows significant effort made by the authors to improve the manuscript.

My remaining issues focus on the clarity of writing, the methods description, and the statistical discussion of curvature versus paleoaccumulation. Now that the paper is refocused on the small-scale accumulation trends, the authors need to more convincingly (ie, statistically) show that paleoaccumulation and surface curvature are related if that is to be one of the major claims of the work. There are a few necessary changes before publication, but most comments below highlight opportunities to improve clarity that would simply increase the impact of the work.

**Specific Comments:**

The methods description is at times unclear, and seems to overcomplicate what is a fairly simple method. Most of my confusion is the result of equations (2), (3), and (4):

1) The basic premise of the paper is that an ice flow model can be used to compute the strain thinning rate: the single unknown required to convert observed, dated layer thicknesses to accumulation rates. According to my read, this makes equations (2) and (4) totally unnecessary. You use the model to infer tau, plug tau into equation (3), and compute a. All the equation manipulation is superfluous – a_o, x_o, and tau are all self-consistent model defined values, just plug in tau, don't substitute a_o and x_o into (3) for tau when a_o and x_o are ultimately a product of tau. It makes the whole computation seem much more complicated to the reader than it is.

2) Using the framework currently provided, it is not clear that equations (2) and (3) can be meaningfully equated. The point the authors make here is that there is residual error between the modeled isochron positions and the observed isochron positions. For a constant time period (defined in $\Delta\chi$) compared across model and observations, the depth to the layers in the model and observations will be different, making the bounds of integration in (2) and (3) different, changing the value for $\int \tau^{-1}dz$ , and preventing the equality presented in (4). Alternatively, for a depth range held constant, you would be comparing different mean ages between model and data, introducing another complication to your interpretations.

Additional confusion then came from lines 158-161, where you say you have not yet incorporated any temporal variation into your calculation. But, equation three calculates the spatial pattern of accumulation rate for discrete layer packages in time, giving you four separate time steps of computed accumulation

rates. What new information does R(t) provide, beyond the accumulation rates that you've derived from the data? This needs to be made much clearer for the reader.

My final comment lies with figure 6 and the argument made about spatial correlations in accumulation rate and surface curvature. I still struggle to get the critical information out of figure 6 – the high accumulation rate in the band across the center of the image does appear (somewhat) well correlated, but elsewhere there appears to be significant disagreement between the surface curvature and the accumulation rates, especially in radar lines not contained in the central grid. To argue that these are correlated, the authors need to show (either as a figure that directly relates detrended accumulation rate to surface curvature, or through some statistical metric) the strength of the correlation. It is apparent from your text that the correlation is not as strong statistically as it might be visually (they say in lines 389 and 390 that there is no good functional fit between CPWD and paleoaccumulation), but it is important for the sake of accuracy that this be emphasized to the reader. That paleoaccumulation and surface curvature may appear related in some locations, but also appear unrelated in others, and that is apparent in the authors inability to define a functional relationship between the two parameters. Right now, the conclusions feel a bit overstated.

With those issues addressed, my only remaining technical corrections are focused on the writing, which are not crucial changes before publication. I do think that there are opportunities to improve readability, but those changes are at the discretion of the authors.

**Technical Corrections:**

| Line # | Comment |
|---|---|
| 70-72 | The specifics of the 1D assumption (and how you know when and where it is violated) need to be discussed before you make this statement. A reader who isn't already familiar with your methods would have little idea what this means. |
| 75-87 | This paragraph isn't methods. It should come in the introduction. |
| 100 | "… add a number of shallower and deeper isochrones …" seems strange to list both "shallower" and "deeper". You just added isochrones. |
| 101 | You should remove the in prep citation. You are not relying on conclusions from another study in this sentence, and citing papers that have not gone through peer review should only be done in extraordinary circumstances. |
| 106-133 | I think much of what is said here is not necessary for the paper, and actually makes everything much more confusing. From a reader perspective, it would be much simpler for you to define tau and how it comes from the model (and also its units or how to interpret its values, you quote it as a value of 0.9 later and it took me a while to figure out what that meant), and go from there. |

| 114 | This definition leads to significant confusion for me later, because you calculate the spatial pattern for several different time periods (despite saying here that you have a temporally invariant spatial pattern). I think you are better off saying "for each time interval between dated isochrons, we assume a fixed spatial pattern for accumulation rate (a_bar[x]), that can vary in time at frequencies higher than those captured by the radar data (R[t])." |
|---|---|
| | I'm not sure you need this definition at all, it seems like the primary conclusion of the paper is that you computed an accumulation rate field through time a(t,x), using discrete t values chosen based on pickable isochrons. But regardless of how you decide to phrase this, know that it takes a lot of mental energy right now to keep track of exactly what your terms mean, and when things are spatially invariant and when they are temporally invariant. |
| 154 | Define how and why you chose the 5km threshold here. This should come after the explanation that shows up at 162-168, where you explain where 1D models work and where they don't. |
| 187+191 | In this paragraph you simultaneously say that the ECMWF ERA40 reproduces and doesn't reproduce the accumulation. You need to be more precise in your wording here, it makes for a confusing paragraph. |
| 208 | Might help to change the section heading for 2.5 to clearly indicate you are looking at spatial trends. |
| 221-226 | This is all methods, not results. |
| 262-270 | I find this paragraph and methods description very confusing. You have a proxy for average Holocene accumulation already, it is your $\Delta\chi = 0 - 10{,}000a$. What are you doing here that is better than the radar derived product? |
| 301 | Given that tau is a term unique to the model, you need to define earlier what a value of 0.9 would mean. That, after strain thinning, the layer is at 90% of its original thickness. Otherwise, the reader has little idea how to interpret this sentence. You also probably shouldn't use the word "pollute", it feels imprecise. |
| 380 | Get rid of one of the drifts. "… and mass drift drift transport." |
| 390 | This is the point where you must convince the reader that the surface topography and accumulation are well correlated. You should plot one against the other, to show the reader the strength of the correlation. The map presented in figure 6 can be used to help motivate why some regions might be more well correlated than others, but to prove a baseline correlation you should just plot those two variables together. |